# ADAPTIVE EXPANSION FOR HYPERGRAPH LEARNING

## ABSTRACT

Hypergraph, with its powerful ability to capture higher-order complex relationships, has attracted substantial attention recently. Consequently, an increasing number of hypergraph neural networks (HyGNNs) have emerged to model the high-order relationships among nodes and hyperedges. In general, most HyGNNs leverage typical expansion methods, such as clique expansion (CE), to convert hypergraphs into graphs for representation learning. However, they still face the following limitations in hypergraph expansion: (i) Some expansion methods expand hypergraphs in a straightforward manner, resulting in information loss and redundancy; (ii) Most expansion methods often employ fixed edge weights while ignoring the fact that nodes having similar attribute features within the same hyperedge are more likely to be connected compared with nodes with dissimilar features. In light of these challenges, we design a novel CE-based **Ad**aptive **E**xpansion method called **AdE** to expand hypergraphs into weighted graphs that preserve the higher-order hypergraph structure information. Specifically, we first introduce a Global Simulation Network to pick two representative nodes for symbolizing each hyperedge in an adaptive manner. We then connect the rest of the nodes within the same hyperedge to the corresponding selected nodes. Instead of leveraging the fixed edge weights, we further design a distance-aware kernel function to dynamically adjust the edge weights to make sure that node pairs having similar attribute features within the corresponding hyperedge are more likely to be connected with large weights. After obtaining the adaptive weighted graphs, we employ graph neural networks to model the rich relationships among nodes for downstream tasks. Extensive theoretical justifications and empirical experiments over five benchmark hypergraph datasets demonstrate that AdE has excellent rationality, generalization, and effectiveness compared to classic expansion models.

## 1 INTRODUCTION

Hypergraphs, unlike pairwise relationships in traditional graphs, introduce hyperedges to connect multiple nodes, allowing for the representation of complex relationships. This concept has gained significant interest in various domains, such as social networks, community detection, and recommendation systems (Li et al., 2013; Zhang et al., 2022; Ma et al., 2022; Xia et al., 2022; An et al., 2021). To leverage the benefits of hypergraphs, a bunch of hypergraph neural networks (HyGNNs) (Jiang et al., 2019; Yi & Park, 2020; Zhang et al., 2020; Wei et al., 2022; Liao et al., 2021) have been proposed to model the rich connectivity patterns within hypergraphs effectively. Generally speaking, most HyGNNs methods primarily leverage typical expansion methods, e.g., clique expansion (CE) (Sun et al., 2008), star expansion (SE) (Agarwal et al., 2006), and line expansion (LE) (Yang et al., 2022), to convert hypergraphs into graphs, e.g., bipartite graphs (Chien et al., 2022; Xue et al., 2021) or weighted graphs (Feng et al., 2019), and further employ neural networks to model the complex structure within hypergraphs. Specifically, the CE technique substitutes hyperedges with cliques that edges connect every pair of nodes. For instance, HyperGCN (Yadati et al., 2019) utilizes a CE-based method to transfer the hypergraph structure into a weighted graph structure. The LE method, proposed by LEGNN (Yang et al., 2022), aims to construct a graph where vertices in the converted graph are made up of node-hyperedge pairs, as illustrated in Figure 1. If two vertices share the same node or hyperedge in the hypergraph, the two vertices are connected.

Although existing HyGNNs with classic expansion methods achieve excellent performance in modeling the complex relationships among nodes and hyperedges, these works (Yadati et al., 2019; Feng et al., 2019; Zhang et al., 2020; Yang et al., 2022) still face the following limitations in hypergraph

expansion: (i) Some existing methods, such as CE-based methods, convert hypergraphs into graphs based on the hypergraph structure in a relatively straightforward manner, often resulting in information loss or redundancy. For instance, the classic CE method connects all node pairs within the same hyperedge and makes it a fully connected subgraph, which will bring the redundant information in the converted graph (Sun et al., 2008). HyperGCN, another classic CE-based method, proposes to pick two nodes to symbolize the corresponding hyperedge and further connect the rest of the nodes with the selected nodes (Yadati et al., 2019). However, the selected nodes are not representative enough of the corresponding hyperedge, which causes certain information loss. (ii) Most methods employ the fixed weights on

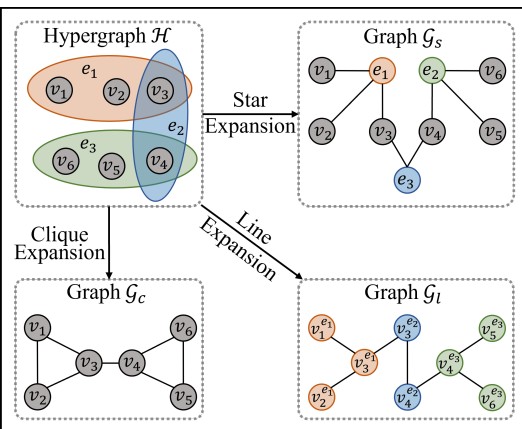

Figure 1: Illustration of classic hypergraph expansion methods.

each edge when expanding hypergraphs into graphs while ignoring that nodes with similar attribute features are more likely to be connected with higher weights during the expansion. For instance, HyperSAGE converts hypergraphs into bipartite graphs where the edge weights are uniformly assigned and further utilizes the hyperedge-level and node-level message-passing strategy to propagate the information among nodes in the converted bipartite graphs (Arya et al., 2020). Yet, this work ignores that nodes with similar attributes within the same hyperedge may have stronger connections during hypergraph expansion.

To handle the above challenges, we propose a novel CE-based hypergraph expansion method called **Ad**aptive **E**xpansion (**AdE**) that expands hypergraphs into weighted graphs, which depicts the higher-order complex relationships among nodes. Specifically, to handle the first challenge, instead of connecting all nodes within the same hyperedge indiscriminately, we design a novel Global Simulation Network (GSi-Net) to select two nodes for symbolizing each hyperedge adaptively. In particular, we first employ a pooling layer to obtain the global representations of the attribute features, followed by a simulation network to learn the importance of each feature dimension, and further obtain the adaptive weight matrix for the attribute features. After obtaining the scaled attribute feature with the adaptive weight matrix, we select two representative nodes for each hyperedge dynamically. To address the second challenge, with the selected nodes for each hyperedge, we design a novel distance-aware kernel function to learn the edge weights to make sure that the edge between two nodes with similar attribute features will have a higher weight during hypergraph conversion. Last, the weighted graph can be fed to any graph neural networks (GNNs) for representation learning. To conclude, this work makes the following contribution:

- *Novelty:* We design a novel expansion method called AdE, including a GSi-Net and a distance-aware kernel function to expand hypergraphs into weighted graphs adaptively over different downstream tasks, which is the first work that learns to find the optimal graph structures adaptively during hypergraph expansion.

- *Generalization:* Our model is designed as a general expansion method that expands hypergraphs into weighted graphs, enabling powerful GNNs to effortlessly and seamlessly study hypergraphs. Theoretic justification also demonstrates the generalization of AdE.

- *Effectiveness:* Theoretic justification and empirical experiments over five benchmark hypergraph datasets demonstrate the effectiveness of our model.

## 2  RELATED WORKS

**Hypergraph Expansions in Hypergraph Neural Networks.** Unlike graphs where the edge connects two nodes, hypergraphs allow hyperedge to connect an arbitrary number of nodes which can depict high-order complex relationships. Most hypergraph neural networks (HyGNNs) first convert hypergraphs to (weighted) graphs via classic expansion methods, i.e., clique expansion (CE) (Sun et al., 2008), star expansion (SE) (Agarwal et al., 2006), line expansion (LE) (Yang et al., 2022) and its variants (Yadati et al., 2019; Feng et al., 2019; Zhou et al., 2006) and further feed the con-

verted graph to neural networks for representation learning. Clique expansion, one of the classic expansions, replaces hyperedges with cliques in which every node of pairs within the corresponding hyperedge is connected. For instance, recent studies, i.e., HyperGCN (Yadati et al., 2019) and HGNN Feng et al. (2019) propose updated CE-based methods to convert hypergraphs into weighted graphs and further learn the hypergraph representations for node classification tasks through graph neural networks, e.g, GCN (Kipf & Welling, 2017) and GAT (Veličković et al., 2018). Star expansion, another classic expansion, creates a set of nodes that represent hyperedges and further connects each new node with nodes that are in the corresponding hyperedge. The graph constructed by SE forms a bipartite graph. Several methods (Chien et al., 2022; Arya et al., 2020) are inspired by SE that treats the hypergraph as a bipartite graph and builds message-passing functions on hypergraphs (Yang et al., 2022). For example, UniGNN (Huang & Yang, 2021) generalizes GNNs (Kipf & Welling, 2017; Veličković et al., 2018; Xu et al., 2019) to hypergraph, and AllSet further unifies a whole class of two-stage models with multiset functions (Chien et al., 2022). Line expansion (LE) (Yang et al., 2022) constructs a graph with a new set of nodes where each node represents a node-hyperedge pair in the original hypergraph, and any two new nodes are connected if they share the same node or hyperedge on the hypergraph. LEGNN leverages LE to build new graphs and further learn the node embeddings through GNNs. However, these methods fail to preserve high-order information within hypergraphs concisely and may lead to undesired losses in hypergraph conversion. Inspired by existing HyGNNs, this work proposes an adaptive hypergraph expansion to expand hypergraphs into weighted graphs to depict the higher-order relationships among nodes.

**Graph Neural Networks.** Graph neural networks (GNNs) (Hamilton et al., 2017), considering both the node features and the graph structure, have become the state-of-the-art approach to learning graph representations. We would like to introduce several popular GNNs: Graph Convolution Network (GCN) Kipf & Welling (2017) implements a layer-wise propagation rule to learn the node embedding; Graph Attention Network (GAT) (Veličković et al., 2018) employs attention mechanisms to measure the importance of neighboring nodes when aggregating features; Graph Isomorphism Network (GIN) (Xu et al., 2019) is designed to utilize the Weisfeiler-Lehman Isomorphism Test Weisfeiler & Leman (1968) for neighbor aggregation to enhance the capacity of GNNs in distinguishing different graph structures. To leverage the powerful GNNs to learn the complex interaction in hypergraphs, inspired by existing works (e.g., HyperGCN and HGNN), we feed the weighted graph via our designed adaptive hypergraph expansion method to GNNs for downstream tasks.

## 3 PRELIMINARY

**Definition 3.1. Hypergraph.** Given a hypergraph $\mathcal{H} = (\mathcal{V}, \mathcal{E}, \mathcal{X})$, $\mathcal{V}$ is the set of nodes with size $N = |\mathcal{V}|$, $\mathcal{E}$ is the set of hyperedges with size $M = |\mathcal{E}|$, and $\mathcal{X}$ is the attribute feature set. Unlike the pair-wise edge in a graph that only connects two vertices, each hyperedge represents a higher-order interaction among a set of nodes. A hypergraph can be represented by an incidence matrix $\mathbf{H} \in \mathbb{R}^{N \times M}$, where $\mathbf{H}_{v,e} = 1$ if $v \in e$; otherwise, $\mathbf{H}_{v,e} = 0$. Here node $v \in \mathcal{V}$ and hyperedge $e \in \mathcal{E}$. Besides, we use $d(v) = \sum_{e \in \mathcal{E}} \mathbf{H}_{v,e}$ and $d(e) = \sum_{v \in \mathcal{V}} \mathbf{H}_{v,e}$ to denote the degrees of node and hyperedge, respectively.

**Definition 3.2. Graph Neural Networks (GNNs).** Most GNNs (Wu et al., 2019; Wang et al., 2019; Schlichtkrull et al., 2018; Yun et al., 2019; Liu et al., 2021) follow the neighbor aggregation operation in the messaging passing framework. Specifically, each node receives and aggregates the messages from the neighbor nodes recursively in multiple layers. For instance, the propagation rule of GNNs is formulated as follows:

$$\mathbf{Z}_{i,:}^{(l+1)} = \mathcal{M}(\mathbf{Z}_{i,:}^{l}, \{\mathbf{Z}_{j,:}^{l} : v_j \in \mathcal{N}_i\}; W^{(l+1)}), \tag{1}$$

where $\mathbf{Z}^{(l+1)}$ is the $(l+1)$-th layer embeddings, $\mathcal{N}_i$ is the neighbors of node $v_i$, and $\mathcal{M}(\cdot; W^{(l+1)})$ is the $(l+1)$-th message passing function with parameters $W$. Note that our framework is applicable to any GNNs, e.g., GCN (Kipf & Welling, 2017) and GAT (Veličković et al., 2018).

**Problem 1.** *Given a hypergraph $\mathcal{H} = (\mathcal{V}, \mathcal{E}, \mathcal{X})$ with ground-truth labels $Y$, the objective is to design a hypergraph expansion method to convert hypergraph to a weighted graph $\mathcal{G}_a$ and further employ GNNs with a mapping function $f_\phi : \mathcal{V} \to \mathbb{R}^d$ (with parameter $\phi$) to project each node $v_i \in \mathcal{V}$ to a $d$-dimensional embedding for downstream tasks with $Y$.*

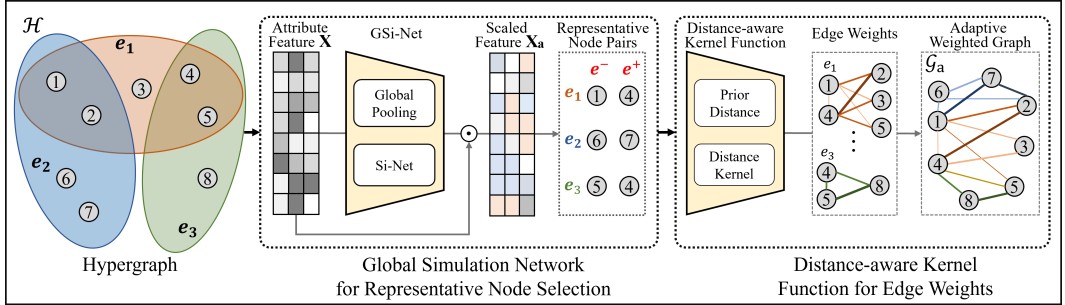

Figure 2: The overall framework of AdE: given a hypergraph $\mathcal{H} = (\mathcal{V}, \mathcal{E}, \mathcal{X})$ with attribute feature $\mathbf{X}$, (i) AdE first feeds the attribute feature $\mathbf{X}$ into a global pooling layer, and leverages simulation network (Si-Net) to learn the importance of each feature dimension, further obtaining the weight matrix $W_g$. Afterward, AdE scales the attribute feature $\mathbf{X}_a$ with the learnable weight matrix $W_g$ and identifies a representative node pair $(v_{e^-}, v_{e^+})$ for each hyperedge $e$. (ii) with representative node pairs, AdE designs a distance-aware kernel function with prior distance information to adaptively learn the edge weights and further constructs an adaptive weighted graph $\mathcal{G}_a$.

## 4 METHODOLOGY

In this section, we present the details of our new **Ad**aptive **E**xpansion method called AdE, which includes the following two key stages: (i) the **G**lobal **Si**mulation **Net**work (GSi-Net) for adaptive representative node selection and (ii) the distance-aware kernel function to learn the weights among node pairs dynamically.

### 4.1 GSI-NET FOR ADAPTIVE REPRESENTATIVE NODE SELECTION

Previous works (Yadati et al., 2019; Feng et al., 2019; Bai et al., 2021; Chien et al., 2022) primarily convert hypergraphs into graphs via some expansion methods (e.g., CE) and further leverage neural networks to learn the complex structure within hypergraphs. The classic CE method is not concise enough and may lead to information redundancy as every pair of nodes within the corresponding hyperedges is connected. To handle this, HyperGCN proposes an updated CE-based expansion method that employs hypergraph Laplacian with mediators (Chan & Liang, 2020) to expand hypergraphs $\mathcal{H}$ to graphs $\mathcal{G}$ by learning a non-linear function over the real-valued signal $\mathbf{S}$ (Yadati et al., 2019). Specifically, HyperGCN first computes a real-valued signal $\mathbf{S} \in \mathbb{R}^N$ where $\mathbf{S} = \mathbf{X} \cdot W_r$, with random feature weight $W_r$. With the signal $\mathbf{S}$, for each hyperedge $e$, HyperGCN selects two nodes $(v_{e^+}, v_{e^-})$ based on the following rule: $(v_{e^+}, v_{e^-}) = \arg\max_{v_i, v_j \in e} |\mathbf{S}_i - \mathbf{S}_j|$. The aforementioned node selection approach is based on the existing works (Zhang et al., 2017; Zhou et al., 2006), which show that two nodes within the largest distance within the hyperedge can represent the information within the hyperedge to a large extent. Then HyperGCN connects these two nodes, and all other nodes in the hyperedge with two representative nodes $v_{e^+}$, and $v_{e^-}$, respectively. Afterward, HyperGCN obtains the weighted graph, where each edge is assigned a fixed weight $\frac{1}{2|e|-3}$, for further representation learning.

However, these existing methods still face the following limitations: (i) Due to the uncertainty of the random matrix for the real-value signal $\mathbf{S}$, the selected representative nodes for each hyperedge are biased and are not representative enough for the corresponding hyperedge. (ii) The edge weights among node pairs in $\mathcal{G}$ are fixed, which ignores the fact that nodes with similar attribute features are more likely to be connected. To this end, we first design a **G**lobal **S**imulation **Net**work (GSi-Net) to select the most representative node pair for each hyperedge adaptively. Furthermore, we design a distance-aware kernel function to dynamically adjust the weights on edges in a weighted graph $\mathcal{G}_a$.

**Global Simulation Network.** Due to the limitation of the existing methods for representative node selection based on the attribute features, we propose to learn a flexible and informative attribute matrix that can be utilized to identify the most characteristic node pair for the hyperedge dynamically. Specifically, we first employ a pooling layer, i.e., global mean pooling, to obtain the global level representation of the attribute features $\mathbf{X}_g \in \mathbb{R}^{1 \times b}$, where $\mathbf{X}_g = \frac{1}{N} \sum_{i=0}^{N} \mathbf{X}_{i,:}$ and $b$ is the feature

dimension. Afterward, we design a simulation network (Si-Net) to adaptively learn the importance of each feature dimension, which is formulated as follows:

$$W_{\mathrm{g}} = \sigma(W_2 \cdot \mathrm{ReLU}(W_1 \cdot \mathbf{X}_{\mathrm{g}})), \tag{2}$$

where $W_{\mathrm{g}} \in \mathbb{R}^{1 \times b}$ is the weight matrix and $\sigma$ is the sigmoid activation function. Moreover, we further scale the attribute feature $\mathbf{X}_{\mathrm{a}}$ with the weight matrix $W_{\mathrm{g}}$ of the attribute feature, where $\mathbf{X}_{\mathrm{a}} = \mathbf{X} \odot W_{\mathrm{g}}$. By dynamically learning the importance of each feature dimension, we hope that our scaled attribute feature $\mathbf{X}_{\mathrm{a}}$ is informative for selecting the excellent node pair for representing the corresponding hyperedge.

**Representative Node Selection.** Instead of using the original attribute feature with random noise as the real-value signal $\mathbf{S}$ in HyperGCN, we employ the sum of scaled attribute feature $\mathbf{X}_{\mathrm{a}}$ among feature dimensions as the signal $\mathbf{S} = \sum_{k=1}^{b} \mathbf{X}_{\mathrm{a},(:,k)}$, where $\mathbf{X}_{\mathrm{a},(:,k)}$ is the column $k$ of the scaled attribute feature $\mathbf{X}_{\mathrm{a}}$ and $\mathbf{S} \in \mathbb{R}^{N}$, to adaptively select the excellent node pair to symbolize the corresponding hyperedge, as introduced in HyperGCN (Yadati et al., 2019). In specific, for each hyperedge $e \in \mathcal{E}$, we pick two nodes $(v_{e^-}, v_{e^+})$ to symbolize the hyperedge $e$ based on the following rule: $(v_{e^-}, v_{e^+}) = \arg\max_{v_i, v_j \in e} |\mathbf{S}_i - \mathbf{S}_j|$. Mention that, similar to existing works (Chan & Liang, 2020; Chan et al., 2018; Louis, 2015), if multiple node pairs satisfy the rule above, we randomly select one node pair as the representation of hyperedge $e$. Subsequently, the remaining nodes within hyperedge $e$, denoted as $\mathcal{V}_m^e = \{v_m | v_m \neq v_{e^-}, v_m \neq v_{e^+}, v_m \in e\}$, are mediators. Furthermore, each mediator in $\mathcal{V}_m^e$ will connect with the representative nodes $v_{e^-}$ and $v_{e^+}$, respectively, and we then obtain the weighted graph $\mathcal{G}_{\mathrm{a}}$ from hypergraph $\mathcal{H}$. The edge set in the weighted graph $\mathcal{G}_{\mathrm{a}}$ for each hyperedge $e$ is denoted as $\mathcal{E}_e$, where $\mathcal{E}_e = \{\{v_{e^-}, v_{e^+}\}, \{v_m, v_{e^-}\}, \{v_m, v_{e^+}\} | v_m \in \mathcal{V}_m^e\}$. Unlike existing hypergraph expansion methods, i.e., CE, SE, and LE, that expand identical graph structures for all downstream tasks, our model AdE learns to generate optimal graph structures via GSi-Net by selecting different representative node pairs adaptively for different downstream tasks. To intuitively demonstrate the differences between our model AdE and hypergraph expansion method in HyperGCN, we provide detailed comparison and illustration in Appendix B.

## 4.2 DISTANCE-AWARE KERNEL FOR EDGE WEIGHT

As we mentioned, most existing methods assign the fixed edge weight to node pairs in the weighted graph $\mathcal{G}_{\mathrm{a}}$ (Yadati et al., 2019; Dong et al., 2020; Arya et al., 2020). However, it ignores the fact that node pairs with similar attribute features should have higher edge weights as they are more likely to be connected compared with nodes with dissimilar attribute features. For instance, HyperGCN fixes the edge weight as $\frac{1}{2|e|-3}$, but does not consider that nodes within the same hyperedge may exhibit different behaviors. Let's take an intuitive example of social media. Given a social media hypergraph, where nodes are users and hyperedges describe a group of users who share the same interests. Assume a group of users is interested in photography, and two individuals focus on automotive photography while the rest are interested in landscape photography. In this case, when expanding the hypergraph to a social media graph, the fixed edge weight among these node pairs introduced by HyperGCN is inappropriate enough as these two individuals should get closer compared with the rest of the users within the group. In light of this, we design a distance-aware kernel function to learn the edge weight in an adaptive manner. Specifically, we first pre-compute the distance matrix $\mathcal{U}$ (i.e., Euclidean distance) among nodes based on the original attribute feature $\mathbf{X}$, where $\mathcal{U}_{i,j} = ||\mathbf{X}_{i,:} - \mathbf{X}_{j,:}||_2$. We hope that the prior information $\mathcal{U}$ can guide the learning process of edge weights by considering the influence of attribute features. Afterward, we design a learnable kernel function to adjust the edge weight in $\mathcal{G}_{\mathrm{a}}$, which is formulated as follows:

$$\mathcal{W}_{i,j} = \exp(-\frac{1}{b} \sum_{d=1}^{b} \frac{\mathcal{U}_{i,j}(\mathbf{X}_{\mathrm{a},(i,d)} - \mathbf{X}_{\mathrm{a},(j,d)})^2}{\theta_d^2}), \tag{3}$$

where $\theta$ is a learnable matrix, $b$ is the dimension of $\mathbf{X}_{\mathrm{a}}$, and $\mathbf{X}_{\mathrm{a},(i,d)}$ is the $d$-th element of the node $v_i$ in the scaled attribute feature $\mathbf{X}_{\mathrm{a}}$. Unlike existing hypergraph expansion methods that assign fixed weight to edges regardless of the downstream tasks, our novel distance-aware kernel function learns the edge weights dynamically. The edge weight between node pairs will be different if we handle different downstream tasks, which is common in real-world scenarios. Next, we would like to provide theoretical justifications for our designed distance-aware kernel function.

**Proposition 1.** *Our designed kernel function $\mathcal{W}$ is distance-aware.*

*Proof Sketch.* In Eq. 3, the kernel function $\mathcal{W}$ is non-negative, as it is governed by the exponential function, which is strictly positive. On the one hand, the function is aware of the feature distance between the node pair $(v_i, v_j)$, as it calculates the distance in the scaled attribute space via $(\mathbf{X}_{a,(i,d)} - \mathbf{X}_{a,(j,d)})^2$. The summation across the squared differences effectively captures the Euclidean distance between node $v_i$ and node $v_j$. On the other hand, $\mathcal{U}_{i,j}$, as the pre-computed distance matrix in the original attribute feature space, also guides the learning process of the edge weights. To conclude, our designed kernel function $\mathcal{W}$ is a distance-aware function. $\square$

**Proposition 2.** *The kernel function $\mathcal{W}$ learns to assign higher edge weights for node pairs with similar attribute features while smaller ones for less similar node pairs.*

*Proof Sketch.* The kernel function $\mathcal{W}$ in Eq. 3 can be broken into four parts: the distance between node pairs in the scaled feature space $\mathbf{X}_a$, the prior distance matrix $\mathcal{U}$, the learning parameters $\theta$, and the exponential function. Firstly, for a node pair $(v_i, v_j)$, we calculate the distance among each dimension via $\Delta^2_{d,(i,j)} = (\mathbf{X}_{a,(i,d)} - \mathbf{X}_{a,(j,d)})^2$ to quantify how close it is between the node pair. Besides, we employ the pre-computed distance matrix in the original feature space as the prior information to guide the learning process of the edge weights. Moreover, we introduce a learnable parameter $\theta$ to control the sensitivity of the node feature similarity across $b$ dimensions. Therefore, these three parts can be denoted as $\mathcal{W}_{i,j} = \exp(-\frac{1}{b}\Gamma_{i,j})$, where $\Gamma_{i,j} = \sum_{d=1}^{b} \frac{\Delta^2_{d,(i,j)}}{\theta_d^2}\mathcal{U}_{i,j}$. In $\Gamma_{i,j}$, a smaller $\theta_d$ would make the function more sensitive to feature differences in the $d$ dimension, while a larger $\theta_d$ would make it less sensitive. We aim to learn these sensitivities dynamically during training to effectively capture the importance of the node features in each dimension.

Considering the two nodes occupy very similar behaviors, the distance in $\Delta^2_{d,(i,j)}$ and the value of $\mathcal{U}_{i,j}$ are supposed to be smaller. Besides, $\theta_d$ learns to facilitate the similarity measures. In this case, the value of $\Gamma_{i,j}$ will be smaller while the negative $\Gamma_{i,j}$ will be larger, resulting in a larger value of $\mathcal{W}_{i,j}$. Therefore, these two similar nodes would be assigned larger edge weights. Otherwise, when their attribute features are less similar, the value of $\Gamma_{i,j}$ will be larger while the negative $\Gamma_{i,j}$ will be smaller, leading to a smaller value of $\mathcal{W}_{i,j}$. $\square$

The above theoretical justifications show that our kernel function $\mathcal{W}$ is a distance-aware kernel function that adaptively assigns larger edge weights to similar node pairs and lower ones to less similar node pairs. Inspired by exiting work (Chan & Liang, 2020), we standardize the edge weights such that the edge weights corresponding to hyperedge $e$ are sum to 1. Formally, for each edge $\{v_i, v_j\} \in \mathcal{E}_e$, we compute the normalized weight $\bar{\mathcal{W}}^{(e)}_{i,j}$ with respect to hyperedge $e$ as follows:

$$\bar{\mathcal{W}}^{(e)}_{i,j} = \frac{\mathcal{W}_{i,j}}{\sum_{\{v_k,v_g\}\in\mathcal{E}_e}\mathcal{W}_{k,g}}. \tag{4}$$

By the aforementioned steps, we further obtain the adaptive weighted graph $\mathcal{G}_a = (\mathcal{V}, \mathcal{E}_a, \mathcal{X}_a)$ with the adaptive adjacency matrix $A_a$, where $A_{a,(i,j)} = \sum_{e\in\mathcal{E}} \mathbb{I}[\{v_i, v_j\} \in \mathcal{E}_e]\bar{\mathcal{W}}^{(e)}_{i,j}$. To intuitively demonstrate our proposed method AdE, we provide an example about AdE implement in real-world scenarios in Appendix J

### 4.3 REPRESENTATION LEARNING

After obtaining the adaptive weighted graph $\mathcal{G}_a = (\mathcal{V}, \mathcal{E}_a, \mathcal{X}_a)$, inspired by UniGNN (Huang & Yang, 2021) that leverages GNNs over the converted graphs, we also employ powerful GNNs as graph encoders to learn the node embeddings over the adaptive weighted graph $\mathcal{G}_a$. Here, we take a two-layer GCN (Kipf & Welling, 2017) as an example, denoted as AdE-GCN. The propagation rule of AdE-GCN to generate node embeddings is defined as follows:

$$\mathbf{Z} = A_a\text{ReLU}(A_a\mathbf{X}_aW^{(1)})W^{(2)}, \tag{5}$$

where $A_a$ and $\mathbf{X}_a$ are the weighted adjacency matrix and scaled attribute feature matrix in $\mathcal{G}_a$, respectively. Meanwhile, $W^{(1)}$ and $W^{(2)}$ are the learnable weight matrices for the first layer and second layer, respectively. Note that our model is designed as a general expansion method, which means it can be effortlessly and seamlessly employed in any GNNs.

In this work, we employ the node classification task to evaluate the effectiveness of our designed method. Thus, we feed the node embeddings $\mathbf{Z}$ generated via GCN into the *softmax* function to generate the probability distribution $\mathbf{P}$, and the semi-supervised node classification loss $\mathcal{L}$ is formulated

as $\mathcal{L} = -\sum_{v_i \in \mathcal{V}} \sum_{c \in Y} y_{i,c} \log(\mathbf{P}_{i,c})$, where $y_{i,c}$ is the label of node $v_i$. The pseudo-code of AdE is listed in Appendix Algorithm 1.

Next, we would like to prove that our expansion method expands a more effective graph compared with existing methods, such as HyperGCN.

**Proposition 3.** *Given the same selected nodes* $(v_{e^-}, v_{e^+})$ *for hyperedge e, our model AdE enhances HyperGCN by generating more adaptive weighted graphs.*

**Proposition 4.** *Our AdE is equivalent to weighted clique expansion in 3-uniform hypergraphs.*

The theoretical justification for Proposition 3 and Proposition 4 are provided in Appendix F and Appendix G, respectively.

## 5 EXPERIMENTS

In this section, we first introduce the experimental setup, including the benchmark datasets, baseline methods, and experimental settings. We then compare AdE with various baseline methods to show the effectiveness of AdE. Moreover, ablation studies and embedding visualizations are conducted to show the rationality and effectiveness of AdE. More details about data statistics, baseline settings, and complexity analysis are provided in Appendix C, Appendix D, and Appendix E, respectively. To show the strong applicability of AdE, we conduct additional experiments on AdE for the hyperedge prediction task, and baseline methods about node-degree preserving hypergraph projection in Appendix H and Appendix I, respectively.

### 5.1 EXPERIMENT SETUP

**Benchmark Datasets.** To evaluate the effectiveness of our model, we employ five benchmark hypergraph datasets from (Chien et al., 2022), including two co-authorship networks, i.e., Cora-CA and DBLP, and three co-citation networks, i.e., Cora, Citeseer, and Pubmed. More detailed discussion and data statistics are introduced in Appendix C.

**Baseline methods.** We compare our model with four classic hypergraph expansion methods, including clique expansion (CE) based methods (G1), star expansion (SE) based methods (G2), line expansion (LE) based methods (G3), and Uni-based methods (G4). To fairly compare with these hypergraph expansion methods, we leverage three GNN models, i.e., Graph Convolution Network (GCN) (Kipf & Welling, 2017), Graph Attention Network (GAT) (Veličković et al., 2018), and Graph Isomorphism Network (GIN) (Xu et al., 2019), as backbone models over the converted graphs. Since vanilla GAT and GIN do not consider the edge weights, inspired by this work (Spielman & Srivastava, 2008), we employ a threshold to consider the edge weights during information propagation. Besides, to further comprehensively evaluate the effectiveness of our model, we also conduct experiments on MLP and five HyGNNs benchmark models, including HGNN (Feng et al., 2019), HCHA (Bai et al., 2021), HyperGCN (Yadati et al., 2019), HNHN (Dong et al., 2020), and AllSet (Chien et al., 2022). Details about baseline methods are introduced in Appendix D.

**Experimental Settings.** Our model adopts accuracy as the evaluation metric to evaluate the performance over our mode and baseline methods. We randomly select $50\%$ of data as the training data, and the rest of the data is split evenly between the validation ($25\%$) and testing sets ($25\%$). Additionally, we conduct each method five times with 500 epochs and report the average score with standard deviation (std). All experiments are conducted under the environment of the Ubuntu 22.04.3 OS plus an Intel i9-12900K CPU, two GeForce RTX 3090 Graphics Cards, and 64 GB of RAM. We utilize Adam (Kingma & Ba, 2015) as the optimizer. We define specific ranges to find the optimal hyper-parameters in our model. For instance, the range of hidden dimensions for layers is $\{64, 128, 256, 512\}$, the range of weight decays is $\{0, 0.001, 0.0001, 0.00001\}$, the range of threshold is $\{0, 0.1, 0.15, 0.2, 0.25, 0.3\}$, and the range of learning rate is $\{0.1, 0.01, 0.001, 0.0001\}$. Then, we report the best performance among the optimal hyper-parameters for each method across all benchmark citation hypergraphs.

### 5.2 EXPERIMENT ANALYSIS

**Performance Comparison.** Table 1 shows the accuracy performance among hypergraph expansions and our model over five benchmark hypergraph datasets. The best performance of our model is highlighted in purple, and the best performance among all baseline methods is highlighted in

Table 1: Performance comparison (Mean accuracy % ± std) of hypergraph expansion methods on GNNs for node classification. Purple-shaded numbers indicate the best results of our models, and gray-shaded numbers represent the best results of baseline methods.

| Group | Model | Cora-CA | DBLP | Cora | Citeseer | Pubmed |
|-------|-------|---------|------|------|----------|--------|
| G1 | CE-GCN | 77.99 ± 0.85 | 87.03 ± 0.25 | 77.24 ± 1.32 | 70.44 ± 0.70 | 81.42 ± 0.54 |
| | CE-GAT | 78.05 ± 1.27 | 87.25 ± 0.28 | 76.04 ± 1.56 | 69.30 ± 1.79 | 81.34 ± 0.53 |
| | CE-GIN | 79.26 ± 1.35 | 88.90 ± 0.13 | 76.81 ± 2.30 | 69.70 ± 1.03 | 84.31 ± 0.25 |
| G2 | SE-GCN | 81.01 ± 1.37 | 89.57 ± 0.25 | 79.04 ± 1.32 | 71.03 ± 2.03 | 82.14 ± 0.48 |
| | SE-GAT | 81.14 ± 1.78 | 89.45 ± 0.29 | 78.91 ± 2.65 | 71.14 ± 1.48 | 83.32 ± 0.38 |
| | SE-GIN | 77.82 ± 1.46 | 88.62 ± 0.29 | 75.82 ± 2.36 | 71.79 ± 1.21 | 85.75 ± 0.55 |
| G3 | LE-GCN | 81.71 ± 1.41 | 90.21 ± 0.29 | 77.23 ± 1.28 | 70.80 ± 2.06 | 81.62 ± 0.39 |
| | LE-GAT | 81.68 ± 1.61 | 90.14 ± 0.37 | 76.99 ± 1.53 | 68.55 ± 1.70 | 82.93 ± 0.82 |
| | LE-GIN | 81.74 ± 1.89 | 90.08 ± 0.34 | 76.75 ± 2.02 | 71.59 ± 0.76 | 84.30 ± 0.47 |
| G4 | Uni-GCN | 79.74 ± 1.96 | 89.06 ± 0.95 | 78.86 ± 1.54 | 71.38 ± 0.81 | 85.72 ± 0.53 |
| | Uni-GAT | 78.98 ± 2.47 | 86.66 ± 1.22 | 77.92 ± 2.03 | 71.30 ± 0.82 | 85.33 ± 0.27 |
| | Uni-GIN | 77.59 ± 2.49 | 89.52 ± 1.15 | 76.84 ± 1.25 | 71.55 ± 1.15 | 85.72 ± 0.36 |
| Ours | AdE-GCN | 82.07 ± 0.88 | 90.53 ± 0.46 | 80.74 ± 1.94 | 72.49 ± 0.94 | 87.68 ± 0.54 |
| | AdE-GAT | 80.79 ± 1.32 | 90.41 ± 0.16 | 80.71 ± 0.74 | 71.57 ± 1.42 | 84.26 ± 0.25 |
| | AdE-GIN | 81.71 ± 1.25 | 90.86 ± 0.16 | 79.11 ± 2.70 | 70.34 ± 1.56 | 87.88 ± 0.43 |

Table 2: Performance comparison (Mean accuracy % ± std) of baseline methods for node classification. Purple-shaded numbers indicate the best results and gray-shaded numbers represent the runner-up performance.

| Model | Cora-CA | DBLP | Cora | Citeseer | Pubmed |
|-------|---------|------|------|----------|--------|
| MLP | 71.34 ± 1.33 | 84.79 ± 0.18 | 71.34 ± 1.33 | 69.32 ± 1.54 | 86.34 ± 0.34 |
| HGNN | 81.32 ± 2.23 | 90.52 ± 0.14 | 79.59 ± 2.59 | 71.62 ± 0.82 | 82.02 ± 0.39 |
| HCHA | 80.81 ± 1.92 | 90.52 ± 0.26 | 79.62 ± 2.41 | 71.24 ± 1.36 | 82.17 ± 0.42 |
| HyperGCN | 79.08 ± 1.53 | 89.54 ± 2.93 | 78.39 ± 2.03 | 70.39 ± 1.09 | 82.43 ± 1.88 |
| HNHN | 73.41 ± 1.82 | 87.77 ± 0.22 | 72.38 ± 2.04 | 69.82 ± 1.93 | 80.72 ± 0.53 |
| AllSet | 81.00 ± 4.10 | 91.34 ± 0.27 | 78.18 ± 1.53 | 71.55 ± 1.41 | 86.87 ± 1.13 |
| Ours | 82.07 ± 0.88 | 90.86 ± 0.16 | 80.74 ± 1.94 | 72.49 ± 0.94 | 87.88 ± 0.43 |

gray. According to Table 1, we make the following conclusions: (i) Merely leveraging the classic CE method to expand hypergraphs is not sufficient to depict the complex high-order relationships among hypergraphs, as all model performances in G1 have the worst performance compared with other groups. (ii) Other hypergraph expansion methods achieve relatively satisfactory performance over different hypergraphs, showing their advanced ability over some hypergraphs. For instance, LE-GIN shows excellent performance over Cora-CA, DBLP, and SE-based methods show better performance over Cora, Citeseer, and Pubmed hypergaph datasets. (iii) Compared with all methods, our AdE-based methods outperform all baseline methods, which shows the strong effectiveness of AdE. Table 2 shows the accuracy performance among MLP, five HyGNNs methods, and our AdE. The best performance is highlighted in purple, and the runner-up performance is highlighted in gray. Based on Table 2, we find that: (i) all HyGNNs models outperform the feature-based method MLP, showing that hypergraph structure enhances the performance for node classification tasks to a large extent. (ii) CE-based methods, i.e., HNHN, HGNN, HCHA, and HyperGCN show comparable performance over Cora-CA, Cora, and Citeseer, while SE-based method AllSet, outperforms these CE-based methods on DBLP and Pubmed. (iii) AdE method outperforms all HyperGNNs on Cora-CA, Cora, Citeseer, and Pubmed, and shows competitive performance over DBLP, which again shows the superiority of our model.

A1: AdE removes GSi-Net and Kernel Function; A2: AdE removes GSi-Net; A3: AdE removes Kernel Function.

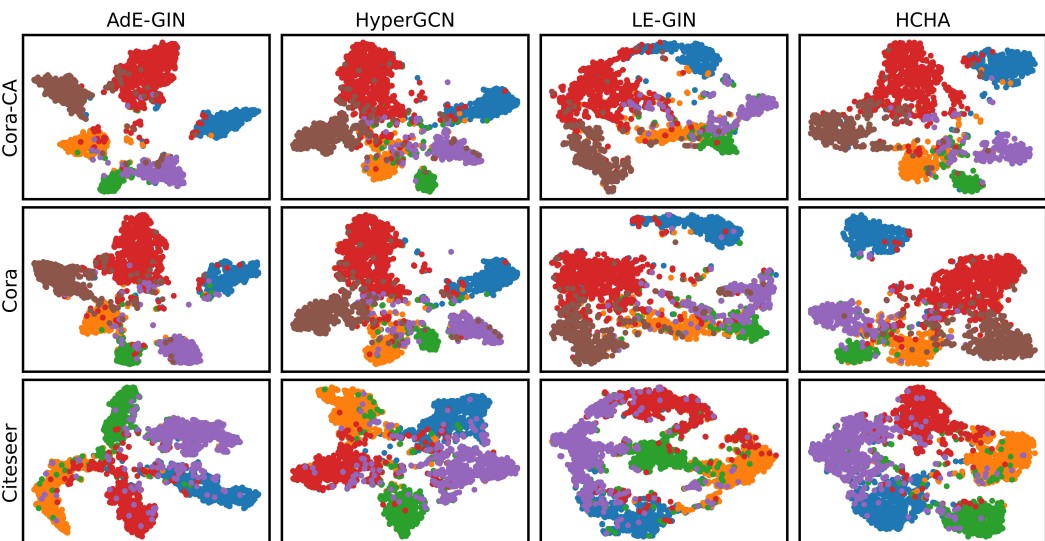

Figure 3: Performances of different model variants over DBLP, Cora, Citeseer, and Pumbed.

**Ablation Study.** To show the effectiveness of each component in our framework, we conduct a set of ablation experiments over four benchmark hypergraph datasets for node classification tasks and further analyze the contribution of each component in our framework, i.e., our GSi-Net and Kernel function (A1), GSi-Net (A2), kernel function (A3), by removing it separately, as illustrated in Figure 3. First, we remove the GSi-Net and kernel function from our model (A1), which means we employ classic clique expansion to obtain the graph and feed the converted graph to GNNs for representation learning. We conclude that our adaptive expansion method is effective enough as the performance of A1 drops significantly on all four hypergraph datasets. Afterward, we remove GSi-Net from our model, which means we utilize the distance-aware kernel function to assign weights to the graph obtained via classic CE. The performance of A2 decreases obviously in all four datasets, showing the effectiveness of the GSi-Net module. Moreover, we remove the kernel function from our model, which means we merely employ the GSi-Net to obtain the graph while assigning the fixed edge weights. The decline of A3 shows that the kernel function has contributed to our model.

**Embedding Visualization.** To further examine the effectiveness of our model intuitively, we render the embeddings of three citation datasets generated by AdE-GIN, HyperGCN, LE-GIN, and HCHA in Figure 4. Each unique color represents the embeddings corresponding to a specific class. According to Figure 4, our model shows more distinct boundaries and smaller overlapping areas compared with other baseline methods, which again demonstrates the effectiveness of our expansion on hypergraph representation learning for node classification tasks.

Figure 4: Embeddings visualization for AdE-GIN, HyperGCN, LE-GIN, and HCHA over citation hypergraphs, including Cora-CA, Cora, and Citeseer.

## 6 CONCLUSION

In this paper, we introduce a novel CE-based adaptive expansion method called AdE to address the limitations of existing hypergraph expansion methods. We first introduce a novel global simulation network called GSi-Net to choose two representative nodes for each hyperedge in an adaptive manner to symbolize each hyperedge. Then we devise a distance-aware kernel function that dynamically

adjusts the edge weights to ensure that nodes with similar attribute features within the corresponding hyperedge are more likely to be connected. Afterward, we leverage graph neural networks to model the complex interaction among nodes for downstream classification tasks. We also provide extensive theoretical justifications and experiments over five benchmark hypergraphs to demonstrate that AdE is rational, general, and effective, compared to classic expansion methods.

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

# APPENDIX

## A    ALGORITHM

---

**Algorithm 1:** Training Procedure of AdE

---

**Data:** Hypergraph $\mathcal{H}$, GSi-Net, distance-aware kernel function, and GNN encoder $f(\cdot)$.
**Result:** The adaptive weighted graph $\mathcal{G}_a$ and trained GNN encoder.

1 Pre-compute the distance matrix $\mathcal{U}$.
2 **for** *each epoch $t$* **do**
3     Feed $\mathcal{H}$ into GSi-Net to generate the scaled attribute feature $\mathbf{X}_a$.
4     Select representative nodes $(v_{e^-}, v_{e^+})$ based on $\mathbf{X}_a$ to symbolize the hyperedge $e$.
5     Compute the distance-aware edge weight $\mathcal{W}$ via the learnable kernel function in Eq. 3.
6     Normalize the edge weight via Eq. 4.
7     Obtain the adaptive weighted graph $\mathcal{G}_a = (\mathcal{V}, \mathcal{E}_a, \mathcal{X}_a)$.
8     Feed weighted graph $\mathcal{G}_a$ into GNN encoder $f(\cdot)$ to generate node embedding $\mathbf{Z}$ via Eq. 5.
9     Optimize the GNN encoder $f(\cdot)$, the parameters in GSi-Net, and the parameters in distance-aware kernel function by minimizing the cross-entropy loss $\mathcal{L}$.

---

## B    COMPARISON BETWEEN EXPANSION IN HYPERGCN AND ADE

Figure 5 shows the comparison between our method AdE with the hypergraph expansion in HyperGCN. Given a hyperedge $e$, HyperGCN selects two nodes following the rule: $(i_e, j_e) = \arg\max_{i,j \in e} |\mathbf{S}_i - \mathbf{S}_j|$, where $\mathbf{S} = \mathbf{X} \cdot W_r$ with random weight matrix $W_r$, and $\mathbf{S} \in \mathbb{R}^N$. It then connects $(i_e, j_e)$, and each other node in hyperedge $e$ to $i_e$ and $i_j$, respectively, with a fixed weight of $\frac{1}{2|e|-3}$, forming a subgraph on the left. However, our proposed method AdE leverages GSi-Net to learn the weight matrix $W_g = \sigma(W_2 \cdot \text{ReLU}(W_1, \mathbf{X}_g))$ to generate the signal $\mathbf{S} = \sum_{d=1}^{b} \mathbf{X}_{:,d} \odot W_g$, where $\mathbf{S} \in \mathbb{R}^N$. AdE then finds a pair of representative nodes $(v_{e^+}, v_{e^-}) = \arg\max_{v_i, v_j \in e} |\mathbf{S}_i - \mathbf{S}_j|$. Subsequently, AdE connects $(v_{e^+}, v_{e^-})$, and each node $v_m \in e, v_m \notin \{v_{e^+}, v_{e^-}\}$ with $v_{e^+}$ and $v_{e^+}$, respectively, with weights learned via an adaptive kernel function $\mathcal{W}_{i,j}$.

Next, we would like to conclude the process of AdE as follows:

1. For each hyperedge $e \in \mathcal{E}$, AdE selects two representative nodes $(v_{e^+}, v_{e^-}) = \arg\max_{v_i, v_j} |\mathbf{S}_i - \mathbf{S}_j|$, where $\mathbf{S} = \sum_{k=1}^{b} \mathbf{X}_{:,k} \cdot \sigma(W_2 \cdot \text{ReLU}(W_1 \cdot \mathbf{X}_g))$. The rest of nodes in hyperedge $e$, i.e., $\mathcal{V}_m^e = \{v_m | v_m \neq v_{e^+}, v_m \neq v_{e^-}, v_m \in e\}$ are mediators.

2. For each hyperedge $e \in \mathcal{E}$, we connect each mediator with two representative nodes $v_{e^+}$ and $v_{e^-}$, respectively, and further obtain the edge set $\mathcal{E}_e = \{\{v_{e^+}, v_{e^-}\}, \{v_{e^+}, v_m\}, \{v_{e^-}, v_m\} | v_m \in \mathcal{V}_m^e\}$.

3. We compute edge weight in each edge set $\mathcal{E}_e$ via our designed learnable kernel function $\mathcal{W}_{i,j}$ in Eq. 3, and normalize these edge weights with respect to hyperedges, $\bar{\mathcal{W}}_{i,j}^{(e)} = \frac{\mathcal{W}_{i,j}}{\sum_{v_k, v_j \in \mathcal{E}_e} \mathcal{W}_{k,g}}$.

By the aforementioned steps, we obtain the weighted graph with the adaptive adjacency matrix $A_a = \sum_{e \in \mathcal{E}} \mathbb{I}[\{v_i, v_j\} \in \mathcal{E}_e] \bar{\mathcal{W}}_{i,j}^{(e)}$.

## C    DATA DESCRIPTION

To evaluate the effectiveness of our model, we employ five benchmark hypergraph datasets adapted from (Yadati et al., 2019): the coauthorship networks, i.e., Cora-CA and DBLP; the cocitation networks, i.e., Cora, Citeseer, and Pubmed. In both coauthor hypergraph datasets, documents co-authored by an author are connected via one hyperedge. In three cocitation hypergraph datasets, all documents referenced by a document are connected by a hyperedge. Table 3 lists the statistics of five benchmark hypergraph datasets.

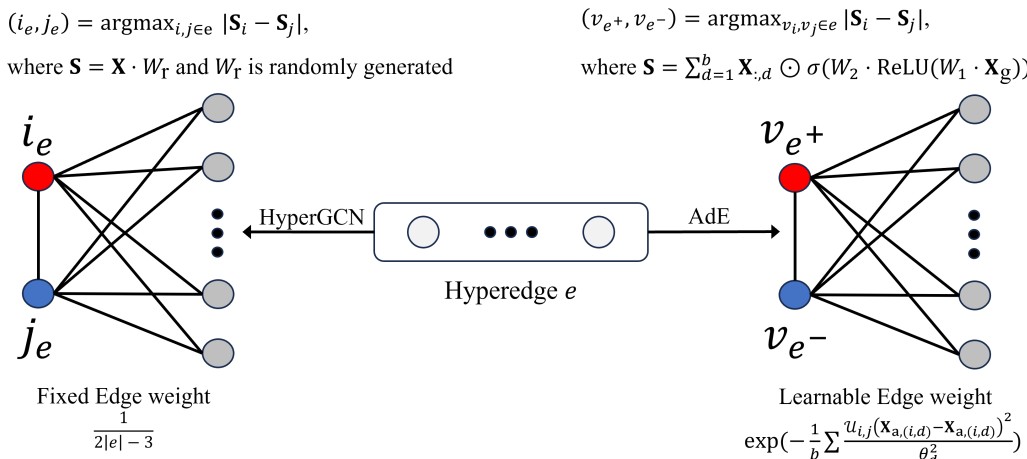

Figure 5: Comparison between hypergraph expansion in HyperGCN and AdE.

Table 3: The statistics of benchmark hypergraph datasets for node classification tasks.

|  | Cora-CA | DBLP | Cora | Citeseer | Pubmed |
|---|---|---|---|---|---|
| # nodes, $N$ | 2,708 | 41,302 | 2,708 | 3,312 | 19,717 |
| # hyperedges, $M$ | 1,072 | 22,363 | 1,579 | 1,079 | 7,963 |
| # features | 1,433 | 1,425 | 1,433 | 3,703 | 500 |
| # class | 7 | 6 | 7 | 6 | 3 |
| avg. $d(e)$ | 4.28 | 4.45 | 3.03 | 3.20 | 4.35 |
| avg. $d(v)$ | 1.69 | 2.41 | 1.77 | 1.04 | 1.76 |
| node homophily | 0.79 | 0.88 | 0.84 | 0.78 | 0.79 |
| hyperedge homophily | 0.88 | 0.93 | 0.86 | 0.83 | 0.88 |

## D  BASELINE SETTINGS

We compare our model with four baseline hypergraph expansion methods, including clique expansion (CE) based methods (G1), star expansion (SE) based methods (G2), line expansion (LE) based methods (G3), and uni-based methods (G4). To fairly compare with baseline hypergraph expansion methods, we leverage three GNN models, i.e., Graph Convolution Network (GCN) (Kipf & Welling, 2017), Graph Attention Network (GAT) (Veličković et al., 2018), and Graph Isomorphism Network (GIN) (Xu et al., 2019), as backbone models. Next, we would like to introduce these baseline methods in each group in detail.

*G1 CE-based methods:* We implement the clique expansion method (Sun et al., 2008) that connects every pair of nodes within each hyperedge. Like the previous work (Chien et al., 2022), we assign edge weights based on the degree of hyperedges connecting the node pair to obtain the weighted graph. Besides, we further apply (row-wise) normalization on the adjacency matrix to ensure stable and efficient feature propagation. We feed the weighted graph to two-layer GNNs, i.e., GCN, GAT, and GIN, to learn the node embeddings for node classification.

*G2 SE-based methods:* We reproduce the star expansion method (Agarwal et al., 2006). SE method creates a star-like structure, i.e., introduces a set of new node, each representing a hyperedge, and aggregate the new node attribute features from the nodes within its corresponding hyperedge. Likewise, the new nodes are connected to these nodes within the hypergraph. We employ two-layer GNNs, i.e., GCN, GAT, and GIN, on the converted graph to generate the node embeddings.

*G3 LE-based methods:* We implement the line expansion method proposed by (Yang et al., 2022). LE method constructs an entirely new graph by creating a set of nodes for each incident node-hyperedge pair in the hypergraph, and two nodes are connected when they share the same node or hyperedge in the original hypergraph. Similarly, we feed the new graph into two-layer GCN, GAT, and GIN for node classification.

Table 4: Accuracy performance comparison (Mean accuracy % ± std) of weighted clique expansion (CE) with distance-aware kernel function $\mathcal{W}$ and adaptive expansion (AdE) without GSi-Net on backbone GNNs for node classification.

| Dataset | Expansion | GCN | GAT | GIN |
|---------|-----------|-----|-----|-----|
| Cora | CE+kernel | 79.28 ± 1.95 | 78.26 ± 2.07 | 77.74 ± 0.94 |
| | AdE\GSi-NET | 79.35 ± 1.04 | 78.35 ± 1.84 | 77.59 ± 1.01 |
| Citeseer | CE+kernel | 71.98 ± 0.98 | 70.31 ± 1.18 | 69.94 ± 1.67 |
| | AdE\GSi-NET | 71.42 ± 0.97 | 70.42 ± 1.32 | 69.80 ± 1.80 |

*G4 Uni-based methods:* We reproduce uni-based expansion (Huang & Yang, 2021), which constructs a bipartite graph with two sets of new nodes, one for nodes in the hypergraph and another for hyperedges. Then, it leverages the pooling function to pass node attribute features to hyperedges and utilizes GNNs to propagate hyperedge attribute features back to new nodes as the final node embeddings. We follow the settings from their source code to reproduce the uni-based expansion.

Besides, to further exhaustively evaluate the effectiveness of our model, we conduct experiments on MLP and five HyGNNs, including HGNN (Feng et al., 2019), HCHA (Bai et al., 2021), HyperGCN (Yadati et al., 2019), HNHN (Dong et al., 2020), and AllSet (Chien et al., 2022). MLP primarily focused on encoding the feature of node attributes while ignoring the graph structures. We employ two-layer MLP on attribute features to generate node representations for classification. HGNN (Feng et al., 2019) employs the convolution operation using truncated Chebyshev polynomials to generate node representations. HCHA (Bai et al., 2021) introduces hypergraph attention and utilizes neural networks to study the node embeddings. HyperGCN leverages hypergraph Laplacian to convert hypergraphs into weighted graphs and further feeds the weighted graph into GCN to learn node embeddings. HNHN (Dong et al., 2020) extends HyperGCN with nonlinear activation functions combined with a normalization schema that adjusts the importance of hyperedges and nodes. AllSet (Chien et al., 2022) unifies a whole class of two-stage message-passing models with multisite functions. We adopt the implementation of HGNN, HCHA, and HNHN from the Pytorch Geometric Library (PyG) (Fey & Lenssen, 2019) and exactly follow the settings of HyperGCN and Allset from their source code to reproduce the experimental results.

# E  COMPLEXITY ANALYSIS

In this section, we discuss the efficiency of AdE in terms of time and space complexity. *Time complexity:* The time complexity for feeding the node feature matrix into GSi-Net is linear to $\mathcal{O}(N)$, where $N$ is the size of nodes. Then, selecting hyperedge representative node pairs takes $\mathcal{O}(M)$ where $M$ is the size of hyperedges. The time complexity to compute the elements in distance matrix $\mathcal{U}$ and generate edges with weights takes $\mathcal{O}(E)$, where $E = \sum_{e \in \mathcal{E}} d(e)$. Since both $\mathcal{O}(M)$ and $\mathcal{O}(N)$ are dominated by $\mathcal{O}(E)$, the total time complexity of AdE for each round is $\mathcal{O}(E)$. *Space complexity:* The space to store the scaled node attribute feature matrix $\mathbf{X}_a$ takes $\mathcal{O}(Nd)$, where $d$ is the size of feature dimensions. Since we adopt a sparse representation to store the adjacency matrix $A_a$, the space should be $\mathcal{O}(E)$. For the pre-compute distance matrix $\mathcal{U}$, instead of storing it in full, we compute the specific elements in the edge construction process, which means it takes a constant space. Moreover, the total space complexity of storing the parameter matrix of GSi-Net and the weight matrix in distance-aware kernel function is $\mathcal{O}(2bh + b) = \mathcal{O}(2bh)$, where $h$ is the hidden dimension. So, the total space complexity of AdE is $\mathcal{O}(Nd + E + 2bh)$, which is linear to the size of nodes $N$.

# F  PROOF OF PROPOSITION 3

**Proposition 3.** *Given the same selected nodes $(v_{e^-}, v_{e^+})$ for hyperedge $e$, our model AdE enhances HyperGCN by generating more adaptive weighted graphs.*

*Proof Sketch.* Given the same selected nodes $(v_{e^-}, v_{e^+})$ for hyperedge $e$, we first analyze the edge weights assigned by HyperGCN. HyperGCN fixes the edge weights for all node pairs within the

Table 5: Performance comparison (Mean accuracy % ± std ) of CEGCN, HyperGCN, and AdE over Cora, Cora-CA, and Citesser datasets for hyperedge prediction. Purple-shaded numbers indicate the best results of our models, and gray-shaded numbers represent the best results of baseline methods.

| Model | Pooling | Cora | Cora-CA | Citeseer |
|---|---|---|---|---|
| CEGCN | Mean | 70.38 ± 2.87 | 53.54 ± 2.06 | 74.50 ± 1.48 |
| | Sum | 79.27 ± 0.36 | 60.38 ± 2.87 | 74.50 ± 1.48 |
| HyperGCN | Mean | 81.16 ± 1.51 | 55.51 ± 3.14 | 77.24 ± 1.92 |
| | Sum | 81.01 ± 0.99 | 61.32 ± 2.71 | 76.51 ± 2.82 |
| AdE | Mean | 84.33 ± 0.99 | 56.09 ± 1.57 | 78.73 ± 1.98 |
| | Sum | 85.53 ± 0.81 | 61.47 ± 1.63 | 77.97 ± 2.09 |

same hyperedge. In specific, as the size of hyperedge $e$ is $2|e| - 3$ in the converted graph, the edge weights on any node pairs assigned by HyperGCN is $(2|e| - 3)^{-1}$.

As proven in Proposition 2, our kernel function $\mathcal{W}$ in Eq. 3 assigns higher edge weights for node pairs with similar attribute features while smaller edge weights for less similar node pairs. In the worst case, the edge weight $\mathcal{W}_{i,j}$ on any node pair $(v_i, v_j)$ within the hyperedge $e$ remains the same with any arbitrary values $\epsilon^{(e)}$. After normalizing edge weights, the edge weights on any node pairs are $\bar{\mathcal{W}}_{i,j}^{(e)} = (2|e| - 3)^{-1}$, which is exactly the same as HyperGCN. However, our kernel function will assign a higher edge weight if nodes $v_i$ and $v_j$ are similar. This implies that $\mathcal{W}_{i,j} > \epsilon^{(e)}$, and $\bar{\mathcal{W}}_{i,j}^{(e)} > (2|e| - 3)^{-1}$. Therefore, we can conclude that AdE enhances HyperGCN in generating more adaptive weighted graphs when they have the same selected nodes $(v_{e^-}, v_{e^+})$. □

## G  PROOF OF PROPOSITION 4

**Proposition 4.** *Our AdE is equivalent to weighted clique expansion in 3-uniform hypergraphs.*

*Proof Sketch.* Given a hypergraph $\mathcal{H} = (\mathcal{V}, \mathcal{E}, \mathcal{X})$, as mentioned by (Sun et al., 2008), the weighted graph $\mathcal{G}_c$ via CE can be represented by the following adjacency matrix:

$$A_{c,(i,j)} = \sum_{e \in \mathcal{E}} \mathbf{H}_{i,e} \mathbf{H}_{j,e} w_{i,j}, \tag{6}$$

where $\mathbf{H}$ denotes the incident matrix, and $w$ is a weight function that computes the weight of edges under some criteria, e.g., $w_{i,j} = |\{e|e : \{v_i, v_j\} \in e, e \in \mathcal{E}\}|$ (Rodriguez, 2003). Here, we use $\bar{\mathcal{W}}_{i,j}^{(e)}$ in Eq. 4. As we introduced, the adjacency matrix generated by AdE is $A_{a,(i,j)} = \sum_{e \in \mathcal{E}} \mathbb{I}\left[\{v_i, v_j\} \in \mathcal{E}_e\right] \bar{\mathcal{W}}_{i,j}^{(e)}$. Consider a 3-uniform hypergraph where each hyperedge contains three nodes. AdE leverages GSi-Net to choose two nodes as $v_{e+}$ and $v_{e-}$, with the remaining one as the mediator. Edges then are generated to connect any two nodes within the corresponding hyperedge. Since there is only one mediator for each hyperedge in a 3-unfirm hypergraph, AdE will generate the same graph structure. Therefore, the indicator function $\mathbb{I}\left[\{v_i, v_j\} \in \mathcal{E}_e\right] = \mathbf{H}_{i,e} \mathbf{H}_{j,e}$. As the weight matrix $w$ in Eq. 6 can be any form, we can replace $w$ with $\bar{\mathcal{W}}$. Therefore, our designed method AdE is equivalent to the weighted clique expansion in a 3-uniform hypergraph. □

To further justify the Proposition 4, we conduct another set of experiments on CE and AdE among datasets Cora and Citeseer as these two hypergraphs are approximately 3-uniform hypergraphs since the average node degree $d(v)$ is close to 3 (as listed in Table 3). The experimental performance of CE+kernel and AdE without GSi-Net is listed in Table 4. To make it clear, we apply our distance-aware kernel function to CE and remove the GSi-NET in AdE for a fair comparison. According to this table, we can see that both methods have equivalent performance with the same settings over two datasets, which further demonstrates our justification of Proposition 4.

Table 6: Performance comparison (Mean F1 score ± std) between IRMM-P, IRMM-GCN, CE-GCN, and AdE over Cora, Cora-CA, and Citesser. Purple-shaded numbers indicate the best results of our models, and gray-shaded numbers represent the best results of baseline methods.

| Model | Cora | Cora-CA | Citeseer |
|-------|------|---------|----------|
| IRMM-P | 39.66 ± - | N/A | 44.10 ± - |
| IRMM-GCN | 49.13 ± 0.54 | 48.36 ± 0.43 | 51.69 ± 0.71 |
| CE-GCN | 77.24 ± 1.32 | 77.99 ± 0.85 | 70.44 ± 0.70 |
| AdE | 80.74 ± 1.94 | 82.07± 0.88 | 72.49 ± 0.94 |

## H  ADE OVER HYPEREDGE PREDICTION TASK

To further demonstrate the strong applicability of AdE, we apply AdE for the hyperedge link prediction task. We follow the work NHP (Yadati et al., 2020) to conduct hyperedge prediction tasks. Specifically, after we generate a weighted graph via AdE and obtain learned embeddings from graph neural networks, we employ a **hyperlink scoring layer** from NHP:

$$I_e = \sigma(W \cdot g(\{h_v\}_{v \in e}) + b), \tag{7}$$

where $W \in \mathbb{R}^{1 \times d}$ is a learnable weight matrix, $g(\cdot)$ is a pooling function, e.g., mean pooling and sum pooling. As mentioned by NHP, the score $I_e$ for hyperedge $e$ needs to be higher than that for any set of nodes that does not form a hyperedge in the hypergraph. Therefore, the objective function is formulated as follows:

$$\mathcal{L} = \frac{1}{|\mathcal{E}|} \sum_{e \in \mathcal{E}} \Lambda((\frac{1}{|\mathcal{F}|} \sum_{f \in \mathcal{F}} I_f) - I_e), \tag{8}$$

where $\mathcal{F}$ is a set of sampled hyperedges, and $I_f$ is the score of the sampled hyperedge $f \in \mathcal{F}$. Here, $\Lambda(\cdot)$ is the logistic function, i.e., $\Lambda(x) = \log(1 + e^x)$. The loss $\mathcal{L}$ tries to maximize the number of hyperedge scores (in $\mathcal{E}$) that are higher than the average score of the sampled hyperedges in $\mathcal{F}$. A more detailed discussion is provided in NHP (Yadati et al., 2020). For experiments, we follow the same settings in NHP, i.e., sample a set of $|\mathcal{E}|$ hyperedges, denoted as $\mathcal{F}$, such that the average degree of $\mathcal{F}$ approximately equals to the average degree of $\mathcal{E}$, and each newly generated hyperedge does not overlap with the original hyperedge set $\mathcal{E}$. Afterward, we concatenate hyperedge sets $\mathcal{F}$ and $\mathcal{E}$ to obtain a new hypergraph $\mathcal{G}' = (\mathcal{V}, \mathcal{E}', \mathcal{X})$. The hyperedges $\mathcal{E}'$ are randomly divided into training, validation, and test sets, with ratios of 50%, 25%, and 25%, respectively. We feed the new hypergraph with training hyperedges into AdE, graph neural networks, and hyperlink score layer to compute scores for each hyperedge and optimize the model via Eq. 8. Additionally, we conduct each method five times with 500 epochs and report average accuracy with standard deviation.

The result of CEGCN, HyperGCN, and AdE over Cora, Cora-CA, and Citesser datasets is listed in Table 5. We conduct experiments on two pooling functions, i.e., mean and sum. According to Table 5, our method AdE outperforms CEGCN and HyperGCN in all three datasets.

## I  COMPARISON WITH METHOD ABOUT NODE-DEGREE PRESERVING HYPERGRAPH PROJECTION.

To validate whether the node-degree preserving projection methods effectively convert hypergraphs into graphs, we then discuss one existing method, IRMM (Kumar et al., 2020), and conduct experiments with IRMM over Cora, Cora-CA, and Citeseer datasets. Specifically, according to the Algorithm 1 listed in IRMM (Kumar et al., 2020),

1. We first initialize a hyperedge weight matrix $W$.
2. We compute a reduced adjacency matrix via the equation: $A = HW(D_e - I)^{-1}H^T$, and zero out the diagonal in $A$. Here, $H$ is the hypergraph incidence matrix, $D_e$ is the node degree matrix, and $I$ is the identity matrix.

3. The reduced adjacency matrix $A$ is fed into the Louvain function (Blondel et al., 2008) to find clusters.

4. To evaluate with ground truth, we follow the settings discussed in IRMM, i.e., leverage agglomerative clustering (Ding & He, 2002) on top of clusters obtained by the Louvain function.

5. For each hyperedge $e$, we count the number $k_i^{(e)}$ of nodes belong to both hyperedge $e$ and cluster $i$, i.e., $k_i^{(e)} = |e \cap C_i|$, where $C_i$ denotes to cluster $i$.

6. For each hyperedge $e$, we update hyperedge weight matrix with respect to hyperedge $e$ as follows: $W_{e,:} \leftarrow 0.5(W_{e,:} + w'_e)$, where $w'_e = \frac{1}{m}\sum_{i=1}^{c}\frac{1}{k_i^{(e)}+1}(\delta(e) + c)$. Here, $m$ is the number of hyperedges, $\delta(e)$ is the degree of hyperedge $e$, and $c$ is the number of clusters.

7. We repeat steps 2-6 until the hyperedge weight matrix converges.

To evaluate the performance of the hypergraph reduction method IRMM, we feed the generated adjacency matrix $A$ into graph neural networks, e.g., GCN, and train graph neural networks with ground truth labels (node labels). Here, we denote IRNN-GCN for the setting. Mention that IRMM is an unsupervised method. In order to compare the IRMM with existing methods in our work, we propose IRNN-GCN for fair comparisons. The results of IRMM-GCN, CE-GCN, and AdE are listed in Table 6. Besides, we also report the result of IRMM from their original paper (Kumar et al., 2020), denoted as IRMM-P. Mention that IRMM-P does not report the standard deviation of performance and the performance for the Cora-CA dataset. According to Table 6, AdE outperforms the other baselines, including IRMM-GCN and IRMM-P. IRMM variants do not show compatible performance with CE-GCN, and AdE. The possible reason is that IRMM tries to maximize the modularity among the hypergraphs in unsupervised settings. But we do think the node-degree preserving hypergraph projection can be the future work about how to learn effective node-degree preserving hypergraph during hypergraph expansion.

## J  INTUITIVE EXAMPLE ABOUT AdE IMPLEMENTATION

In this section, to make it clearer, we provide an intuitive example about AdE implement in real-world scenarios. Suppose we have a hyperedge $e$ that four users (A, B, C, and D) have interests in photographs. Node A is interested in landscape photography, Node B enjoys automotive and landscape photography, Node C is interested in automotive photography, and Node D likes food photography. Based on the above scenario, classic methods, i.e., clique expansion, will generate an edge set $\mathcal{E}_e$ ={(A, B), (A, C), (A, D), (B, C), (B, D), (C, D)}. However, Nodes A and C do not share any interests, and they are supposed to be disconnected or connected with very small edge weights. By contrast, our AdE can professionally handle real-world scenarios as follows:

1. AdE first selects two representative nodes with the largest distance within the hyperedge. Let us say that nodes B and D are selected as the representative nodes as their styles are very dissimilar, meaning they have the longest distance in the attribute space.

2. We connect the rest of the nodes (i.e., A and C) with the two representative nodes, respectively, and we further obtain the edge set $\mathcal{E}_e$ = {(B, D), (B, A), (B, C), (D, A), (D, C)}. Nodes A and C are not connected in $\mathcal{E}_e$, as A (landscape photography) and C (automotive photography) do not have the same interests.

3. With the edge set, we learn to assign the adaptive distance-aware weights to all edges in $\mathcal{E}_e$. We believe that the edge weight between A and D is less than the edge weight between A and B. With the shared interest (landscape photography) between A and B, the distance between A and B is smaller than the distance between A and D. So, the edge weight between A and D is smaller as our distance-aware kernel function tends to assign smaller weights for dissimilar node pairs.

Therefore, AdE first generates a subgraph for the hyperedge $e$, with the edge set $\mathcal{E}_e$ = {(B, D), (B, A), (B, C), (D, A), (D, C)}. Mention that, as AdE learns to select representative nodes during model training for certain downstream tasks, the built subgraph is also adaptive with updated distance-aware weights.

