# OpenReview forum: "Adaptive Expansion for Hypergraph Learning"
_ICLR.cc/2024/Conference — Submitted to ICLR 2024_

### Official Review · Reviewer_7Lez · 2023-10-31

**Soundness:** 3 good
**Presentation:** 3 good
**Contribution:** 2 fair
**Rating:** 5
**Confidence:** 3

**Summary:**

The paper proposed a new algorithm for clique expansion in hypergraph. Clique expansion in the previous methods have the same edge weight for all edges. In this paper, the author proposed an adaptive algorithm which can calculate the edge weight based on the hyperedge. The proposed model solves two challenges: 1) the previous model directly apply a clique expansion causes information loss/redundancy. 2) the fixed edge weight ignores the potential strong connection in one hyperedge.

The model contains two steps. First, a global simulation network is applied to choose the most representative node pair $(v_{e^-}, v_{e^+})$. Secondly, an kernel function calculates the edge weights(distances) from two nodes in the node pair to other nodes. The new graph thus contains three kinds of edges: $(v_{e^-}, v_{e^+})$, $(v_{e^-}, v_{m})$, $(v_{m}, v_{e^+})$. For validation, the task is a classification task. The loss is applied to both two networks(global simulation network and kernel function) separately, which indicates it's not an end-to-end model.

The performance show that the paper outperform several baselines. The ablation study shows that each module is helpful and combining them together make the performance better.

**Strengths:**

The paper is well written and easy to follow. The challenges are clear, the authors proposed a new model solved it. The Gaussian Kernel and the adaptive edge weight seems to be novel in this field. The proposed model is equivalent to weighted clique expansion in 3-uniform hypergraphs, which somehow guarantee the expression power.

**Weaknesses:**

One concern is that the performance doesn't outperform the baseline too much. For some datasets the proposed model only has a performance gain smaller than 1 std, like Cora-CA in table 1 and 2, Cora in table 2.

Also, the proposed model is mainly based on HyperGCN. But other models, such as AllSet, have a completely different scheme. They apply star expansion instead of clique expansion. The reason is that the traditional clique expansion is not able to represent the high order information. Let's say there are four nodes: a,b,c,d. The clique expansion can't tell the difference that whether abc have a strong connection or ab and bc have a connection while ac have no connection. This is one example, it can be more nodes. Although in the proposed model, they have an edge weight to help them classify. But I'm still concerning about if it really helps distinguish between these situations. Since edge weight are pair-wise relation. Hypergraph deals with high-order relation. I wonder if authors have more discussion regarding this.

Meanwhile, since each time the model needs to reconstruct the graph, find the maximum distance in one hypergraph. The time complexity will be a problem if the average $d(e)$ is larger.

Furthermore, some other models using hypergraph contrastive learning performs better than You are AllSet(AllDeepSet) and other models. It seems that the performance is not good enough.

**Questions:**

See weaknesses.

---

> ### Author Response · Authors · 2023-11-16
> **Response to Reviewer 7Lez (1/3)**
>
> We appreciate your valuable comments and would like to respond to your questions and concerns as follows.
>
> *$\textbf{Q1:}$ AdE does not outperform the baseline methods very much.*
>
> $\textbf{A1:}$ We appreciate your observation that the performance of some datasets does not outperform the best baseline very much. We believe even small improvements are significant in this domain. Here, we take two popular methods as examples. Table 1 provides the accuracy performance of **AllSet** [1], the best baseline model, and the corresponding improvements over 15 datasets. Table 1 highlights these improvements that are less than 1.  The improvements in **eleven out of fifteen datasets** are less than 1, and the average improvement over fifteen datasets is 0.425.
>
> ### Table 1: Accuracy performance comparison between AllSet and the best bestline method.
>
> | | Cora | Citeseer| Pubmed | Cora-CA | DBLP-CA| Zoo| 20Newsgroups| mushroom|
> |---|---|---|---|---|---|---|---|---|
> |Allset|78.59|73.08|88.75|83.63|91.53|97.50|81.38|100.00|
> |Best Baseline|80.18|74.12|88.25|84.04|91.69|93.65|81.42|100.00|
> |**Improvement**|$\textbf{-1.59}$|$\textbf{-1.04}$|$\textbf{0.50}$|$\textbf{-0.41}$|$\textbf{-0.16}$|3.85|$\textbf{-0.04}$|0.00|
>
> | |NTU2012|ModelNet40|Yelp|House(1)|Walmart(1)|House(0.6)|Walmart(0.6)|
> |---|---|---|---|---|---|---|---|
> |Allset|88.69|98.20|36.89|69.33|65.46|83.14|78.46|
> |Best Baseline|89.30|98.07|33.04|71.05|62.45|83.27|77.72
> |**Improvement**|$\textbf{-0.61}$|$\textbf{0.13}$|3.85|$\textbf{-1.72}$|3.01|$\textbf{-0.13}$|$\textbf{0.74}$|
>
>
> Table 2 shows the best results of the mean test error (lower is better) in **HyperGCN** [2]. According to Table 2, the improvement in Cora and Citeseer is only 0.04 and 0.05, respectively, and the average improvement of HyperGCN over five datasets is 1.464.
>
> ### Table 2. Mean test error performance comparison between HyperGCN and the best baseline method.
>
> |   | DBLP | Pubmed | Cora-CA | Cora| Citeseer |
> |----------|----------|----------|----------|----------|----------|
> | Best Baseline |25.65|29.41|31.90|32.41|37.40|
> |Best HyperGCN variant|24.09|25.56|30.08|32.37|37.35|
> |**Improvement**|1.56 | 3.85 | 1.82| $\textbf{0.04}$| $\textbf{0.05}$|
>
> From the above two tables, we can find the existing  state-of-the-art methods (e.g., AllSet and HyperGCN) are not satisfactory enough over all datasets, and the relatively high improvement is not always guaranteed over all datasets. Our method, AdE, outperforms most existing baseline methods and the state-of-the-art methods over five benchmark datasets, showing the effectiveness.
>
> [1]  You are AllSet: A Multiset Function Framework for Hypergraph Neural Networks, ICLR 2022.
>
> [2] HyperGCN: A New Method of Training Graph Convolutional Networks on Hypergraphs, NeuIPS 2019.

---

> ### Author Response · Authors · 2023-11-16
> **Response to Reviewer 7Lez (2/3)**
>
> *$\textbf{Q2:}$ How does AdE expansion distinguish the edge differences between different node pairs?*
>
> $\textbf{A2:}$ Thank you for your constructive feedback and we would like to explain that our AdE is aware of the differences between different node pairs. Let us take an intuitive example. As we discuss in Section 4.2, suppose we have a hyperedge $e$  that four users (A, B, C, and D) have interests in photographs.  Node A is interested in landscape photography, Node B enjoys automotive and landscape photography, Node C is interested in automotive photography, and Node D likes food photography. Based on the above scenario:
>
> 1. AdE first selects two representative nodes with the largest distance within the hyperedge. Let us say that nodes B and D are selected as the representative nodes as their styles are very dissimilar, meaning that they have the longest distance in the attribute space.
>
> 2. We connect the rest of the nodes (i.e., A and C) with the two representative nodes, respectively, and we further obtain the edge set $\mathcal{E}_e$ = {(B, D), (B, A), (B, C), (D, A), (D, C)}.  Nodes A and C are not connected in $\mathcal{E}_e$, as A (landscape photography) and C (automotive photography) do not have the same interests, showing that our AdE can handle the situation you mentioned.
>
> 3. With the edge set, we learn to assign the adaptive distance-aware weights to all edges in $\mathcal{E}_e$. We believe that the edge weight between A and D is less than the edge weight between A and B. With the shared interest (landscape photography) between A and B, the distance between A and B is smaller than that between A and D. Furthermore, the edge weight between A and D is less than the edge weight between A and B as our distance-aware kernel function tends to assign larger weights to node pairs with similar attribute features while smaller ones for less similar node pairs (Proposition 2).
>
> Although our weighted graph extracts the pair-wise relationships, we can conclude that AdE is aware of the high-order relationships between hyperedges from the above initiative example. In addition, we also want to demonstrate the statement empirically by comparing it with Allset, which is based on the star expansion (SE). Table 3 shows that AdE outperforms Allset and the best SE-based baseline method in four benchmark datasets, while exhibiting competitive performance in the DBLP dataset.
>
> ### Table 3. The mean accuracy of Allset, best SE-based baseline method, and AdE.
> | | Cora-CA| DBLP|Cora|Citeseer|Pubmed|
> |-----|-----|-----|-----|-----|-----|
> |Allset|81.00|$\textbf{91.34}$|78.18|71.55|86.87|
> |SE|81.14|89.57|79.04|71.79|85.75|
> |AdE|$\textbf{82.07}$|90.86|$\textbf{80.74}$|$\textbf{72.49}$|$\textbf{87.88}$|
>
> Overall, we believe our proposed AdE is able to distinguish the difference between node pairs.  In addition, we believe our AdE is better than existing expansion methods in expressing the high-order relationships within hypergraphs.

---

> ### Author Response · Authors · 2023-11-16
> **Response to Reviewer 7Lez (3/3)**
>
> *$\textbf{Q3:}$ Time complexity for graph reconstruction.*
>
> $\textbf{A3:}$ Thank you for raising the concern about the time complexity of our model. We would like to analyze the time complexity of our model AdE and compare AdE with existing hypergraph expansion methods (i.e., clique expansion, star expansion, and line expansion).
>
> Given the number of nodes $N$ and the number of hyperedges $M$, we suppose the average hyperedge degree $k$ in a hypergraph.  For AdE, we find two representative nodes for each hyperedge $e$ based on the following rule: $(v\_{e^+}, v\_{e^-}) = \arg\max\_{v_i, v_j \in e} |\mathbf{S}\_i - \mathbf{S}\_j|$, where $\mathbf{S}\_i \in \mathbb{R}^1$ is the signal value for node $v_i$. This function takes linear time, i.e., $\mathcal{O}(k)$. Then, we first create edges between representation node pairs $(v_{e^+}, v_{e^-})$. Besides, we connect the rest of the nodes $v_m$ (mediators) with each representative node $v_{e^+}$ or  $v_{e^-}$, respectively, i.e., $(v_{e^+}, v_m)$ and $(v_m, v_{e^-})$. Following these steps, we would obtain $2k-3$ edges for each hyperedge, and the time complexity to compute edge weights within a hyperedge is $\mathcal{O}(2k-3)$. To conclude, the total time complexity of AdE is **$\mathcal{O}(M(k + 2k-3)) = {\mathcal{O}(Mk)}$**.  To make it clearer, we revised the time complexity of AdE in the revision. We need to mention that  $E = \sum_{e\in \mathcal{E}} d(e)$ in the original version is equal to $Mk$ in the current version. Additionally,  the time complexity of HyperGCN in hypergraph expansion is ${\mathcal{O}(Mk)}$.
>
> Clique expansion (CE) generates ${O}(k^2)$ edges for each hyperedge $e$. Therefore, the time complexity of clique expansion is $\mathcal{O}(Mk^2)$. We found out that the time complexity of **CE** is **higher** than that of AdE.
>
> Star expansion (SE)  creates $M$ new nodes and generates $\mathcal{O}(Mk)$ edges during expansion. So the time complexity of SE is $\mathcal{O}(M + Mk) = \mathcal{O}(Mk)$. **AdE and SE** have relatively **comparable** time complexity.
>
> Line expansion (LE) creates $Mk$ new nodes and generates $\mathcal{O}(M^2k^2)$ edges when converting hypergraphs into graphs. The time complexity of LE is $\mathcal{O}(M^2 \times k^2)$,  showing that  **LE** has the **largest** time complexity among all hypergraph expansion methods.
>
> In conclusion, our proposed method AdE is more efficient than CE and LE, and has comparable time complexity with SE and expansion method in HyperGCN.  For clearer clarification, we also list the time complexity of each method in Table 4.
>
> ###  Table 4. The time complexity of hypergraph expansion methods, including AdE, CE, SE, and LE.
>
> |Method | Time Complexity|
> |----|----|
> |AdE| $\mathcal{O}(Mk)$|
> |CE|  $\mathcal{O}(Mk^2)$|
> |SE|  $\mathcal{O}(Mk)$|
> |LE| $\mathcal{O}(M^2k^2)$|
>
> *$\textbf{Q4:}$ Hypergraph contrastive learning seems to outperform state-of-the-arts methods (e.g., AllDeepSet).*
>
> $\textbf{A4:}$ Thank you for your comments. We agree that hypergraph contrastive learning (e.g., HyperGCL [3]) enhances the performance of hypergraph representation learning. However, we believe hypergraph contrastive learning and hypergraph expansion are two important but distinct research fields. Simply comparing hypergraph contrastive learning methods (e.g., HyperGCL) and hypergraph expansion methods (e.g., AdE) is not fair enough.
>
> AdE is a fundamental method that focuses on converting hypergraphs into weighted graphs, similar to CE, LE, and SE. After obtaining the weighted graph, we can employ graph neural networks (GNNs) to learn the node embeddings. Furthermore,  if necessary, we can also perform contrastive learning over the converted weighted graph to enhance the model performance.
>
> We hope that our response has addressed your concerns and that you acknowledge the contribution of our novel model design. If so, we would kindly ask you to adjust your review score . Please let us know if you have any additional comments.
>
>
> [3] Augmentations in hypergraph contrastive learning: Fabricated and generative, NeuIPS 2022.

---

> > ### Comment · Reviewer_7Lez · 2023-11-19
> > **Thank you for your response.**
> >
> > Thank you for the explanation and extra experiments provided by the authors. But I still have some questions.
> >
> > Why the pairwise relation can handle the high-order relationship?
> > 1), In the example, AC is still pairwise relationship. You can handle AC doesn't necessarily mean you can handle higher-order relationship.
> > 2), The method proposed in the paper can only describe that BA and BC are highly related, DA and DC are not highly correlated. No matter whether A and C share similar properties or not, they will always be disconnected. The connection of AC can only be implicitly constructed by B and D. This requires the model to have very subtle edge weights. I'm not sure if the model is able to capture the relation. Meanwhile, regarding the higher-order relation, in your example, a more intuitive example is whether BAC all share one property in this hyperedge or is it BA, BC or AC share different properties. It still needs dedicate edge weight to distinguish.
> >
> > More explanation would be appreciated.
> >
> > Regarding the performance:
> > I assume the model is basically designed on the assumption that the hypergraph is homophily, this is also mentioned by other reviewers. That's the reason why I mention AllSet. The datasets they used contain more complicated(larger edge degree and heterophily) datasets, such as Walmart. I wonder how well the proposed model will work on these datasets. Since for the cora(CO/CA), pubmed, DBLP, and Citeseer, they are all homophily datasets with relatively small average edge degree and average node degree. And the proposed model doesn't outperform the baselines too much. I'm still a little concern about it.

---

> > > ### Author Response · Authors · 2023-11-21
> > > **Response to Follow-up Comments of Reviewer 7Lez (3/3)**
> > >
> > > *$\textbf{Q2:}$ How well will the proposed method work on datasets having larger edge degrees and heterophily?*
> > >
> > > $\textbf{A2:}$ We understand your concerns. Therefore, we also conducted another set of experiments over a benchmark dataset which has a large edge degree and heterophily (small homophily). **Compared with Walmart, the House dataset has a larger hyperedge degree and a smaller homophily**. Please refer to Table 2 for more details. Due to the limited time during the rebuttal period, we primarily conducted experiments over the House dataset and compared the performance of AdE and partial of existing baseline methods (e.g., HGNN, HCHA, HyperGCN, and AllSet). We would like to finish the experiments over more datasets in the later revision, e.g., Walmart.
> > >
> > > **Table 2.** Data statistics: $h_v$: node homophily; $h_e$: hyperedge homophily; $avg. d(v)$: mean node degree; $avg. d(e)$: mean hyperedge degree.
> > > ||Cora|Citeseer|Pubmed|Cora-CA|DBLP|House|Walmart|
> > > |--|--|--|--|--|--|--|--|
> > > |$h_v$| 0.84| 0.78| 0.79| 0.79| 0.88| 0.52| 0.55|
> > > |$h_e$| 0.86 | 0.83| 0.88 | 0.88| 0.93| **0.58**| 0.75|
> > > |$avg. d(v)$| 1.77 | 1.04| 1.76 | 1.69| 2.41| 9.18| 5.18|
> > > |$avg. d(e)$| 3.03 | 3.20| 4.35 | 4.28| 4.45| **34.72**| 6.59|
> > > |||||||||
> > >
> > > \
> > > From Table 3, we can conclude that AllSet outperforms all baseline methods over House (1.0) and House (0.6) datasets. Our method **AdE outperforms AllSet** over these two datasets, **showing the effectiveness of handling the hypergraph datasets having larger hyperedge degrees and small homophily**. The improvement of AdE over House (0.6), is larger (more than double ) than that of House (1.0), showing that our model design based on attribute features benefits the model improvements as more actual features of the House dataset are leveraged during model training with less random noise (60\%) injected into the feature of House dataset.
> > >
> > >
> > >
> > > **Table 3.** Performance comparison among AdE and baseline methods over House (1.0) and House (0.6) datasets.
> > >
> > > | | House (1)  | House (0.6) |
> > > |--|--|--|
> > > | HGNN |66.37 ± 1.82 | 68.85 ± 1.99 |
> > > | HCHA |63.84 ± 3.74 | 70.84 ± 1.70 |
> > > |HyperGCN| 64.71 ± 1.71 ​​|  79.63 ± 2.87  |
> > > |AllSet|  67.18 ± 3.66   |  80.74 ± 3.57   |
> > > |AdE|  **67.80 ± 1.50**   |  **82.06 ± 2.58**   |
> > > |Improvement|  0.62 ± -   |  1.32 ± - |
> > > ||||
> > >
> > > To conclude, as our model learns to select representative nodes adaptively and to adjust the edge weights during model training for downstream tasks, AdE can handle different hypergraph datasets effectively, e.g., large homophily hypergraph, small homophily graph, hypergraph with large mean hyperedge degree, and hypergraph with small hyperedge degree. Due to limited time during the rebuttal session, we are not able to conduct experiments over more datasets, but we will finish these experiments in future revisions.
> > >
> > > \
> > > \
> > > We really appreciate your constructive comments and insightful suggestions.  We hope that our response has addressed all your concerns. If so, we kindly request you to consider adjusting your rating to support this work. Should you have any additional questions, please feel free to let us know!

---

> > > ### Author Response · Authors · 2023-11-22
> > > **Warm Reminder to Reviewer 7Lez**
> > >
> > > Dear Reviewer 7Lez,
> > >
> > > Thank you for your constructive comments and suggestions. With the rebuttal deadline approaching, we kindly remind you to review our response to your follow-up questions. We hope our second-round response has effectively addressed your concerns. If it has, we would greatly appreciate your consideration in adjusting your rating. If you have any additional comments or concerns, please feel free to let us know. We sincerely appreciate your valuable efforts.

---

> > > > ### Comment · Reviewer_7Lez · 2023-11-22
> > > > **Thank you for your response**
> > > >
> > > > Thank you for your reply. I get the point and understand that AdE can somehow capture the high-order relation due to the data and training process, but I'm still not convinced that it can naturally capture the high-order relationship by model design. Meanwhile, the authors provide more experiments on one less homophily dataset, i.e., House(1) and House(0.6). However, the baseline of AllSet is AllDeepSets, not AllSetTransformer. The performance should be comparable instead of "outperform".
> > > >
> > > > Overall, I really appreciate the authors effort during the rebuttal process, but I will not change my score.

---

> ### Author Response · Authors · 2023-11-21
> **Response to Follow-up Comments of Reviewer 7Lez (1/3)**
>
> We appreciate your follow-up comments and we would like to answer your questions as follows:
>
> *$\textbf{Q1:}$ Why the pairwise relation can handle the high-order relationship? 1), In the example, AC is still a pairwise relationship. You can handle AC doesn't necessarily mean you can handle higher-order relationships. 2), The method proposed in the paper can only describe that BA and BC are highly related, DA and DC are not highly correlated. No matter whether A and C share similar properties or not, they will always be disconnected. The connection of AC can only be implicitly constructed by B and D. This requires the model to have very subtle edge weights. I'm not sure if the model is able to capture the relation. Meanwhile, regarding the higher-order relation, in your example, a more intuitive example is whether BAC all share one property in this hyperedge or is it BA, BC, or AC share different properties. It still needs a dedicated edge weight to distinguish.*
>
> $\textbf{A1:}$ We appreciate your constructive comments regarding AdE and we would like to answer your Q1 as follows:
> \
> \
> *Why does the pairwise relation can handle the high-order relationship?*
> \
> \
> $\textbf{A:}$ Our expansion method AdE aims to convert hypergraphs into weighted graphs that can preserve the high-order relation to a large extent. During the expansion via AdE, **a single** pair-wise relation **fails** to extract the high-order relationship within the corresponding hyperedge, but t**he subgraph** that is composed of **multiple pair-wise relationships** among nodes can **depict the high-order relationship** within the hyperedge to some extent.
>
> Let us recap the previous example. Suppose we have a hyperedge $e$  that four users (A, B, C, and D) have interests in photographs.  Here $e$ represents the photography interest. Node A is interested in landscape photography, Node B enjoys automotive and landscape photography, Node C is interested in automotive photography, and Node D likes food photography.  With **AdE**, we will obtain a subgraph with the edge set **$\mathcal{E}_e$ = {(B, D), (B, A), (B, C), (D, A), (D, C)}**. We believe our subgraph can capture the high-order relationships within the hyperedge.
>
> However, some existing hypergraph expansion methods may result in the potential loss of information within the hyperedge. Let us take **star expansion in AllSet** as an example. With the star expansion, we will get a graph with the edge set **$\mathcal{E}_e$ = {(A, e), (B, e), (C, e), (D, e)}**, where $e$ is regarded as a photography node in the converted graph. We find out that although star expansion keeps the relationship between nodes (users) and hyperedges (photography interest), it fails to extract the relationship within the hyperedge. Since nodes of **A, B, C, and D** have the same photograph interest, they **are likely to follow some of them on social media**. These connections are important and useful in graph representation learning while star expansion may **fail to keep this information** inside the hyperedge $e$, **resulting in potential information loss**.
> \
> \
> *No matter whether A and C share similar properties or not, they will always be disconnected. The connection of AC can only be implicitly constructed by B and D.*
> \
> \
> $\textbf{A:}$ In our example, we agree that A and C can only be constructed by B and D. However, we need to mention that the **representative two nodes** are **adaptively selected** via our designed **GSi-Net** based on the following rules during model training:
>
>  $(v\_{e^+}, v\_{e^-})=\arg\max\_{v_i,v_j\in e}|\mathbf{S}_i - \mathbf{S}_j|$, where  \$\mathbf{S}=\sum\_{k=1}^b\mathbf{X}\_{:,k}\cdot
> \sigma(W_2\cdot\text{ReLU}(W_1\cdot\textbf{X}\_\text{g})) \\ \text{and} \\ \\  \textbf{X}\_\text{g}=\frac{1}{N}\sum\_{i=0}^N \mathbf{X}\_{i,:}.
>   \\ \\ \\  \\  \\ \\  \\ (1) $
>
> Here,  $\mathbf{S}\in\mathbb{R}^N$, and $\textbf{X}_\text{g}\in\mathbb{R}^b$ is the global-level representation of attribute features in GSi-Net.
> \
> \
> In other words, **the representative two nodes will not be fixed as B and D during the whole training process, while they are adaptively selected via optimizing the cross-entropy loss for downstream tasks**.  Let us say, it is possible that Node A (landscape photography) and C (automotive photography) in our example can be selected as representative nodes for certain downstream tasks because the $\mathbf{S}$ in the above Eq. 1 is learned to select the representative nodes during the model training for the specific downstream tasks.

---

> ### Author Response · Authors · 2023-11-21
> **Response to Follow-up Comments of Reviewer 7Lez (2/3)**
>
> \
> $\textbf{A:}$ What's more, in order to further demonstrate the rationality and the effectiveness of our representative node selection via GSi-Net, we conduct the **ablation study** to analyze the contribution of **GSi-Net**. Please refer to the following Tabe 1 or Figure 3 in our manuscript for a clearer illustration. From this table, we find out that leveraging GSi-Net (**Improvement via GSi-Net**) to select representative nodes shows an **obvious increase** over seven datasets, which validates the effectiveness of GSi-Net in selecting representative nodes. Besides,  the **improvement via our whole AdE** also shows an **obvious enhancement** over all datasets, demonstrating the effectiveness of our designed model AdE.
> \
> \
> **Table 1**.  **Ablation Study**: Improvement of GSi-Net, distance-aware kernel function, and AdE over seven datasets.
> | | Cora-CA | DBLP | Cora | Citeseer | Pubmed | House (1) | House (0.6) |
> |---|---|---|---|---|---|---|---|
> |HyperGCN | 79.08 | 89.54  | 78.39  | 70.39 | 82.43|64.71 | 79.63 |
> |AdE / Kernel | 81.19 | 90.01 | 79.97 |  70.82 | 87.54 |  66.31 | 80.96|
> |AdE| 82.07 | 90.86 | 80.74 | 72.49  | 87.88 | 67.80|82.06|
> | **Improvement via GSi-Net** | 2.11 | 0.47 | 1.58 | 0.43 | 5.11 | 1.60 | 1.33|
> |**Improvement via Kernel function** | 0.88 | 0.85 | 0.77 | 1.67 | 0.34 | 1.49 | 1.10|
> |**Improvement via our whole AdE** | 2.99 | 1.32 | 2.35 | 2.10| 5.45 | 3.09 | 2.43|
> |||||||||
>
> \
> *This requires the model to have very subtle edge weights. I'm not sure if the model is able to capture the relation.*
> \
> \
> $\textbf{A:}$ We believe our model is able to capture the relationship as we design a novel distance-aware kernel function to learn and adjust the edge weight:
>
> $\mathcal{W}\_{i,j} =\exp(-\frac{1}{b}\sum\_{d=1}^b\frac{ \mathcal{U}\_{i,j}(\mathbf{X}\_{\text{a},({i,d})} - \mathbf{X}\_{\text{a}, (j, d)})^2}{\theta\_d^2}), \\ \\ \text{where} \\ \\ \mathbf{X}\_{\text{a}}=\mathbf{X}\cdot \sigma(W_2\cdot\text{ReLU}(W_1\cdot\textbf{X}\_\text{g})).
>  \\ \\  \\ (2) $
>
> Here $\theta$ is the learnable matrix in our distance-aware kernel function. $\mathcal{U}$ is the pre-computed distance matrix in the original attribute space, guiding our kernel function to assign higher edge weights for similar node pairs while smaller ones for less similar node pairs.
>
> To validate that our distance-aware kernel function is effective in learning the edge weights among node pairs, we also conduct ablation study experiments to analyze the contribution of our kernel function separately. From above Table 1, we realize that the performance **improvement via our designed kernel function** shows **a clear rise** over all datasets, especially over Citeseer and House datasets. We believe the ablation study is a good way to show the effectiveness of each module of AdE, making our model AdE more convincing.
> \
> \
> *Meanwhile, regarding the higher-order relation, in your example, a more intuitive example is whether BAC all share one property in this hyperedge or is it BA, BC, or AC share different properties. It still needs a dedicated edge weight to distinguish.*
> \
> \
> $\textbf{A:}$ As formulated in above Eq. 2, BA, BC, and AC will get different edge weights $\mathcal{W}\_{i,j}$ as each node pair will have different embeddings $(\mathbf{X}\_{\text{a},{i}}, \mathbf{X}\_{\text{a},{j}})$ during the model training and the different pre-computed distance.

---

> ### Author Response · Authors · 2023-11-23
> **Response to Follow-up Comments of Reviewer 7Lez**
>
> We appreciate your effort to understand our model design. We would like to explain your concerns one by one as follows:
>
> *$\textbf{Q1:}$ I get the point and understand that AdE can somehow capture the high-order relationship due to the data and training process, but I'm still not convinced that it can naturally capture the high-order relationship by model design.*
>
> $\textbf{A1:}$ We appreciate your partial recognition that our AdE can capture the high-order relationship during the data and training process. However, we need to mention that our model is designed to capture the high-order relationships within the hypergraphs and we would like to explain it from two aspects:
>
>  1. we design a novel module GSi-Net to adaptively learn the most representative two nodes for each hyperedge based on the specific downstream tasks. In other words, the representative nodes selection process is learned during the model training for different downstream tasks. For example, the representative nodes will be different for node classification tasks and hyperedge link prediction tasks. And the optimal structure of the weighted graph will be learned for different downstream tasks. Therefore, we believe our **GSi-Net is designed to learn the high-order relationships from the representative selection perspective**.
>
>  2. We design a novel distance-aware kernel function to learn the edge weights dynamically. The edge weight between node pairs will also be different if we handle different downstream tasks, which is common in real-world scenarios. **Different edge weights (larger / smaller) show their connections (closer / farther) among nodes with different extents**. Designing an adaptive hypergraph expansion method can handle various downstream tasks in a comprehensive way. **In this way, we believe our designed distance-aware kernel function helps preserve the high-order relationships from the edge weight perspective**.
>
> To conclude, our AdE is designed to learn the high-order relationships within hypergraphs with GSi-Net from the representative selection perspectives and with distance-aware kernel function from the edge weight view. We sincerely hope our explanations can address your concerns.
>
> \
> \
> *$\textbf{Q2:}$ Meanwhile, the authors provide more experiments on one less homophily dataset, i.e., House(1) and House(0.6). However, the baseline of AllSet is AllDeepSets, not AllSetTransformer. The performance should be comparable instead of "outperform.”*
>
>
> $\textbf{A2:}$ Thank you for pointing this out. Due to the limited time during the rebuttal session, we are not able to finish the rest of the experiments (i.e., AllSetTransformer) over the House and Walmart datasets. But we will finish them in future revisions. According to our current experimental results, we believe our AdE outperforms most baseline methods over all benchmark datasets, especially, **AdE outperforms HyperGCN**  (backbone) on **all (seven) benchmark datasets**, with a **2.83 accuracy improvement** on average. Besides, **AdE outperforms the SOTA AllSet** over **six benchmark datasets** and also gains comparable performance on some datasets. Even the SOTA AllSet fails to outperform the best baseline methods over some (eight out of fifteen) datasets. We believe our model is effective and shows promising potential in the future research of hypergraph learning.
>
> Besides, we need to mention the main contribution of AdE. AdE is designed as an adaptive hypergraph expansion framework that the weighted graph that can be generated dynamically over different downstream tasks, which is **the first work that can learn to preserve the high-order relationships within hyperedges and find the optimal graph structures adaptively during hypergraph expansion**. This is a promising research direction and will provide valuable insights for future research in hypergraph learning.
>
> \
> \
> We sincerely hope that our explanation addresses any concerns you may have, and we would greatly appreciate your recognition of the contributions made by our work.

---

### Official Review · Reviewer_1ZMR · 2023-10-31

**Soundness:** 2 fair
**Presentation:** 2 fair
**Contribution:** 2 fair
**Rating:** 6
**Confidence:** 3

**Summary:**

The authors propose a method to project a hypergraph to a corresponding graph where the edge weights are not straightforwardly copied from the hyperedges, but an adaptive approach calculates them. The proposed approach is comprised of two steps - using GS-Net to identify representative nodes, followed by a distance-aware kernel to compute the edge weights.

**Strengths:**

1. The problem space is very interesting. There is very little attention paid to finding innovative ways to convert hypergraphs to graphs by preserving the desired properties.
2. The proposed method outperforms the existing baselines. The ablation study provides an understanding of the contribution of different components.

**Weaknesses:**

1. Existing works like node-degree preserving hypergraph projection [1] are not considered. Clique/star-based expansions are not the right representative of SOTA methods.
2. The idea of using node features to compute edge weights assumes that there is an underlying homophily, which was not well captured by hyperedges (hence, the existence of a hyperedge does not mean the constituent nodes share the same strengthened bond), but the projected graph will capture it better. It requires more justification. (please provide additional justification if I got the idea wrong).
3. The current write-up lacks motivation behind the proposed steps in the method. A short text explaining the intuition behind the reason for selecting representative nodes and follow-up steps would be helpful.

[1] "Hypergraph clustering by iteratively reweighted modularity maximization." Applied Network Science 5.1 (2020): 1-22.

**Questions:**

1. What is the rationale behind having a representative pair of nodes for each hyperedge? Also, provide an explanation on how this approach will cater to hyperedges of varying sizes (it can be 1 to n).
2. How does this approach work on a hypergraph where nodes do not have any attributes? Several domains, such as computational biology, where hypergraphs have extensive applications, but getting node-level features is challenging.
3. Provide more details on the hypergraph datasets, such as their size, average degree distribution, average hyperedge size, etc. This will help explain the result trends.
4. Does the proposed method apply to tasks other than node classification? (such as clustering, hyperedge prediction, etc.)

---

> ### Author Response · Authors · 2023-11-17
> **Response to Reviewer 1ZMR (1/5)**
>
> We appreciate your valuable comments and would like to respond to your questions and concerns as follows.
>
> *$\textbf{Q1:}$ Discussion about methods of node-degree preserving hypergraph projection.*
>
> $\textbf{A1:}$ Thank you for raising a valid point regarding the lack of discussion about works like node-degree preserving hypergraph projection [1, 2]. To validate whether the node-degree preserving projection methods effectively convert hypergraphs into graphs, we then discuss one existing method, IRMM [1], and conduct experiments with IRMM over Cora, Cora-CA, and Citeseer datasets. Note that, due to time limits during the rebuttal session, we are not able to conduct sufficient experiments with IRMM over all datasets, but we will finish them in the final revision. Specifically, according to the Algorithm 1 listed in [1],
>
>   * Step 1: We first initialize a hyperedge weight matrix  $W$.
>
>   * Step 2: We compute a reduced adjacency matrix via the equation:  $A = HW(D_e-I)^{-1}H^T$,  and zero out the diagonal in $A$. Here, $H$ is the hypergraph incidence matrix, $D_e$ is the node degree matrix, and $I$ is the identity matrix.
>
>    * Step 3. The reduced adjacency matrix $A$ is fed into the Louvain function [3] to find clusters.
>
>    * Step 4. To evaluate with ground truth, we follow the settings discussed in IRMM, i.e., leverage agglomerative clustering [4] on top of clusters   obtained by the Louvain function.
>
>  * Step 5.  For each hyperedge $e$, we count the number $k_i^{(e)}$ of nodes belong to both hyperedge $e$ and cluster $i$, i.e., $k_i^{(e)}=|e\cap C_i|$, where $C_i$ denotes to cluster $i$.
>
>  * Step 6. For each hyperedge $e$, we update hyperedge weight matrix with respect to hyperedge $e$ as follows: $W\_{e,:} \leftarrow  0.5(W\_{e,: } + w_{e}')$, where $w_{e}' = \frac{1}{m}\sum_{i=1}^c \frac{1}{k_{i}^{(e)}+1}(\delta(e)+ c)$. Here, $m$ is the number of hyperedges, $\delta(e)$ is the degree of hyperedge $e$, and $c$ is the number of clusters.
>
>  * Step  7. We repeat the steps 2-6 until the hyperedge weight matrix converges.
>
> \
> To evaluate the performance of the hypergraph reduction method IRMM, we feed the generated adjacency matrix $A$ into graph neural networks, e.g., GCN, and train graph neural networks with ground truth labels (node labels). Here, we denote IRNN-GCN for the setting. Mention that IRMM is an unsupervised method. In order to compare the IRMM with existing methods in our work, we propose IRNN-GCN for fair comparisons. The results of IRMM-GCN, CE-GCN, and AdE are listed in Table 1.  Besides, for your convenience, we also report the result of IRMM from their original paper [1], denoted as IRMM-P.
>
> ###  Table 1. F1 score between IRMM-P, IRMM-GCN, CE-GCN, and AdE over Cora, Cora-CA, and Citesser.
>
> | | Cora | Cora-CA| Citeseer|
> |---|---|---|---|
> |IRMM-P |39.66 ± -| N/A | 44.10 ± -|
> |IRMM-GCN | 49.13 ± 0.54|  48.36 ± 0.43 | 51.69 ± 0.71 |
> | CE-GCN |  77.24 ± 1.32 |  77.99 ± 0.85 | 70.44 ± 0.70|
> |AdE| $\textbf{80.74 ± 1.94}$ |$\textbf{82.07± 0.88 }$ |$\textbf{72.49 ± 0.94}$|
>
> Mention that IRMM-P does not report the performance of Cora-CA. According to Table 1, AdE outperforms the other baselines, including IRMM-GCN and IRMM-P. IRMM variants do not show compatible performance with CE-GCN, and AdE. The possible reason is that IRMM tries to maximize the modularity among the hypergraphs in unsupervised settings. But we do think the node-degree preserving hypergraph projection can be the future work about how to learn effective node-degree preserving hypergraph during hypergraph expansion.
>
> \
> What's more, clique expansion, and star expansion may not be the SOTA expansion methods, but they are very classic hypergraph expansion methods. Line expansion [5], which is proposed in  2022, provides a promising direction in the hypergraph expansion field. Besides, HyperGCN proposes a novel method to expand hypergraphs to graphs based on the hypergraph Laplacian with Mediators. Moreover, we also compared AdE with some state-of-the-art hypergraph representation learning methods, such as AllSet [6], which employs star-based expansion methods during hypergraph expansion. Compared with these SOTA methods and classic expansion methods, we believe our model AdE is novel, effective, and generalized.
>
> \
> \
> [1] Hypergraph clustering by iteratively reweighted modularity maximization, Applied Network Science 5.1 (2020): 1-22.
>
> [2] Learning with hypergraphs: Clustering, classification, and embedding, NeurIPS 2006.
>
> [3] Fast Unfolding of Communities in Large Networks, Journal of statistical mechanics: theory and experiment 2008.10 (2008): P10008.
>
> [4] Cluster Merging and Splitting in Hierarchical Clustering Algorithms, ICDM 2002.
>
> [5] Semi-supervised Hypergraph Node Classification on Hypergraph Line Expansion, CIKM 2022.
>
> [6] You are AllSet: A Multiset Function Framework for Hypergraph Neural Networks, ICLR 2022.

---

> ### Author Response · Authors · 2023-11-17
> **Response to Reviewer 1ZMR (2/5)**
>
> *$\textbf{Q2:}$ The idea of using node features to compute edge weights assumes that there is an underlying homophily, which was not well captured by hyperedges (hence, the existence of a hyperedge does not mean the constituent nodes share the same strengthened bond), but the projected graph will capture it better.*
>
> $\textbf{A2:}$ Thanks for you constructive comments. Existing works demonstrate that proper hypergraph construction keeps the homophily [7]. For instance, HyperGCL[7] discusses and analyzes the homophily of existing thirteen hypergraph datasets, which is listed in the following Table 2. We can find out that most of the hypergraphs in Table 2 have relatively high homophily, with 10 out of 13 hypergraphs with a homophily larger than 0.7.
>
> ###  Table 2. The node homophily ($h_v$) and hyperedge homophily ($h_e$) for hypergraph benchmark datasets.
>
> ||Cora|Citeseer|Pubmed|Cora-CA|DBLP-CA|Zoo|20News|Mushroom|NTU2012|ModelNet40|Yelp|House|Walmart|
> |--|--|--|--|--|--|--|--|--|--|--|--|--|--|
> |$h_v$| 0.84| 0.78| 0.79| 0.79| 0.88| 0.35 | 0.49| 0.87| 0.81| 0.88| 0.26| 0.52| 0.55|
> |$h_e$| 0.86 | 0.83| 0.88 | 0.88| 0.93| 0.66| 0.73| 0.96| 0.87|0.92| 0.57| 0.58| 0.75|
>
>
> \
> To make it clear, we would like to provide an intuitive example showing that AdE enhances the ability to keep homophily when expanding hypergraphs into graphs. Suppose we have a hyperedge $e$  that four users (A, B, C, and D) have interests in photographs.  Node A is interested in landscape photography, Node B enjoys automotive and landscape photography, Node C is interested in automotive photography, and Node D likes food photography. Based on the above scenario, classic methods, i.e., clique expansion, will generate an edge set $\mathcal{E}_e$ ={(A, B), (A, C), (A, D), (B, C), (B, D), (C, D)}. However, Nodes A and C do not share any interests, and they are supposed to be disconnected or connected with tiny edge weights. But our AdE can professionally handle real-world scenarios:
>
> 1. AdE first selects two representative nodes with the largest distance within the hyperedge. Let us say that nodes B and D are selected as the representative nodes as their styles are very dissimilar, meaning they have the longest distance in the attribute space.
>
> 2. We connect the rest of the nodes (i.e., A and C) with the two representative nodes, respectively, and we further obtain the edge set $\mathcal{E}_e$ = {(B, D), (B, A), (B, C), (D, A), (D, C)}.  Nodes A and C are not connected in $\mathcal{E}_e$, as A (landscape photography) and C (automotive photography) do not have the same interests.
>
> 3. With the edge set, we learn to assign the adaptive distance-aware weights to all edges in $\mathcal{E}_e$. We believe that the edge weight between A and B is larger than the edge weight between A and D. With the shared interest (landscape photography) between A and B, the similarity between A and B is larger than that between A and D. So, the edge weight between A and B is larger, showing the excellent performance of AdE in keeping the homophily.
>
> Regarding the projected graph that preserves the homophily, we strictly follow the setting of IRMM [1] and conduct a set of experiments on IRMM in Table 1 of the previous response. From that table, the experimental results are not that promising yet. However, exploring methods related to node-degree preserving hypergraph projection for hypergraph expansion could be a promising avenue for our future research.
>
> \
> [7] Augmentations in Hypergraph Contrastive Learning: Fabricated and Generative, NeurIPS 2022.

---

> ### Author Response · Authors · 2023-11-17
> **Response to Reviewer 1ZMR (3/5)**
>
> *$\textbf{Q3:}$ Motivation and further explaination about method design. What is the rationale behind having a representative pair of nodes for each hyperedge?*
>
> $\textbf{Q3:}$ Thank you for your comments. We would like to answer your questions and explain our motivation for our method design step by step. First of all, existing traditional expansion methods struggle to preserve the connection information during expansion. For example, clique expansion (CE) [8] transforms a hypergraph into a graph by connecting every node pair within the hyperedges, resulting in information redundancy. In light of this, a recent work, HyperGCN [9], proposes to leverage Hypergraph Laplacian with mediators to convert hypergraph into a weighted graph. Specifically, for each hyperedge $e$, HyperGCN selects two nodes based on the following rule:
>
> $$
> (v\_{e^+}, v\_{e^-})=\arg\max\_{v_i,v_j\in e}|\mathbf{S}_i - \mathbf{S}_j|,  \\  \text{where}   \\  \mathbf{S}= \mathbf{X} \cdot W_r . \\ \\ \\  \\
>  \\ \\  \\ (1)
> $$
>
> Here, $\textbf{W}\_\text{r}$ is a **random** feature weight and $\mathbf{S}\in\mathcal{R}^N$. HyperGCN is inspired by existing works [4, 7] that **two nodes with the largest distance within the hyperedge can represent the information within the hyperedge to a large extent**. Then, HyperGCN connects all other nodes in the hyperedge with the two representative nodes $v\_{e^+}$ and $v\_{e^-}$ , respectively. Afterward, each edge in the weighted graph is assigned a fixed weight in HyperGCN. Please refer to Figure 2 in [9] for more details.
>
> However, HyperGCN still faces several challenges: (i) the selected two nodes are not representative enough as it $\mathbf{S} =\mathbf{X}\cdot W_r$ is generated via the attribute feature and a $W_r$ is a random matrix. (ii) the edge weight for each node pair is fixed as $\frac{1}{2|e|-3}$. It does not consider the variety of edge weights among node pairs.
>
> In light of this, we first design a GSi-Net module to select the two representative nodes $(v\_{e^+}, v\_{e^-})$ adaptively via function:
>
>  $(v\_{e^+}, v\_{e^-})=\arg\max\_{v_i,v_j\in e}|\mathbf{S}_i - \mathbf{S}_j|$, where $\mathbf{S}=\sum\_{k=1}^b\mathbf{X}\_{:,k}\cdot
> \sigma(W_2\cdot\text{ReLU}(W_1\cdot\textbf{X}\_\text{g})).
>   \\ \\ \\  \\  \\ \\  \\ (2) $
>
> Here,  $\mathbf{S}\in\mathbb{R}^N$, and $\textbf{X}_\text{g}$ is the global level representation of attribute features in GSi-Net. Compared to Eq. 2 in GSi-Net and Eq.1 in HyperGCN, we believe our real-value signal $\mathbf{S}$ (in Eq. 2) learned by GSi-Net is more informative and sufficient to select representative nodes for hyperedges. To handle the second challenge, we design a distance-aware kernel function to adaptively assign the edge weights $\mathcal{W}\_{i,j}$ to edge $(v_i,v_j)$, which is formulated as follows:
>
> $\mathcal{W}\_{i,j} =\exp(-\frac{1}{b}\sum\_{d=1}^b\frac{ \mathcal{U}\_{i,j}(\mathbf{X}\_{\text{a},({i,d})} - \mathbf{X}\_{\text{a}, (j, d)})^2}{\theta\_d^2}). \\ \\ \\  \\
>  \\ \\  \\ (3) $
>
> Compared Eq. 3 to the fixed weight $\frac{1}{2|e|-3}$ in HyperGCN, our adaptive edge weight tends to assign higher edge weights between similar node pairs and lower edge weights between dissimilar node pairs. In order to compare AdE with existing work HyperGCN intuitively, we add a new figure to the revised manuscript. Please refer to Figure 5 for clarification. We hope the above clarification about AdE can facilitate your understanding.
>
> \
> [8] Hypergraph spectral learning for multi-label classification, KDD2008
>
>
> [9] HyperGCN: A New Method of Training Graph Convolutional Networks on Hypergraphs, NeuIPS 2019.

---

> ### Author Response · Authors · 2023-11-17
> **Response to Reviewer 1ZMR (4/5)**
>
> *$\textbf{Q4:}$ Explanation on how this approach will cater to varying-sized hyperedges (it can be 1 to n).*
>
> $\textbf{A4:}$ We will explain how this approach will cater to varying-sized hyperedges one by one:
>
> For Hyperedge with degree 1, it is a self-loop for the node with edge weight 1, AdE will add a self-loop of the node to the graph.
>
> For hyperedge with degree 2, it is a pairwise edge, so this approach will generate an edge between two nodes, with edge weight computed by function $\mathcal{W}$ in Equation 3. Note that hyperedges with a degree less than 3 are uncommon, and all of the benchmark hypergraph datasets used in this paper do not have hyperedge that the average degree is less than 3.
>
> For hyperedge with degree 3, our method connects any pair of nodes within the hyperedge, which forms a triangle. AdE generates exactly the same subgraph structure with clique expansion for hyperedge with degree 3. Our Proposition 4 shows that our proposed method AdE is equivalent to weighted clique expansion in 3-uniform hypergraphs, i.e., hyperedges all have degree 3.
>
> For general cases, i.e., hyperedge degree is larger than 3, AdE converts hypergraphs into a weighted graph as follows:
> 1. For each hyperedge $e\in\mathcal{E}$, AdE selects two representative nodes $(v\_{e^+}, v\_{e^-})=\arg\max\_{v_i,v_j}|\mathbf{S}_i - \mathbf{S}_j|$, where $\mathbf{S}=\sum\_{k=1}^b\mathbf{X}\_{:,k}\cdot
> \sigma(W_2\cdot\text{ReLU}(W_1\cdot\textbf{X}\_\text{g}))$ .
> The rest node in hyperedge $e$, i.e., $\mathcal{V}\_m^e=\\{v_m|v_m\ne v\_{e^+}, v_m\ne v\_{e^-}, v_m\in e\\}$ are mediators.
>
> 2. For each hyperedge $e\in\mathcal{E}$, we connect each mediator with two representative nodes $v\_{e^+}$ and $v\_{e^-}$, respectively, and further obtain the edge set $\mathcal{E}_e =\\{  \\{v\_{e^+}, v\_{e^-}\\},\\{v\_{e^+}, v\_m\\}, \\{v\_{e^-}, v\_m\\}| v_m\in \mathcal{V}_m^e \\}$.
>
> 3. We compute edge weight in each edge set  $\mathcal{E}_e$ via our designed learnable kernel function $\mathcal{W}\_{i,j}$ (Eq. 3), and normalize these edge weights with respect to hyperedges, $\bar{\mathcal{W}}\_{i,j}^{(e)} = \frac{\mathcal{W}\_{i,j}}{\sum\_{\\{v_k,v_j\\}\in\mathcal{E}_e}\mathcal{W}\_{k,g}}$.
>
> By the aforementioned steps, we obtain the weighted graph with the adaptive adjacency matrix $A_\text{a}=\sum\_{e\in\mathcal{E}}\mathbb{I}[\{v_i, v_j\}\in\mathcal{E}_e]  \bar{\mathcal{W}}\_{i,j}^{(e)}$.
>
> \
> To make it clearer, let us take an **intuitive example**. As we discuss in Section 4.2, suppose we have a hyperedge $e$  that four users (A, B, C, and D) have interests in photographs.  Node A is interested in landscape photography, Node B enjoys automotive and landscape photography, Node C is interested in automotive photography, and Node D likes food photography. Based on the above scenario:
>
> 1. AdE first selects two representative nodes with the largest distance within the hyperedge. Let us say that nodes B and D are selected as the representative nodes as their styles are very dissimilar, meaning they have the longest distance in the attribute space.
>
> 2. We connect the rest of the nodes (i.e., A and C) with the two representative nodes, respectively, and we further obtain the edge set $\mathcal{E}_e$ = {(B, D), (B, A), (B, C), (D, A), (D, C)}.  Nodes A and C are not connected in $\mathcal{E}_e$, as A (landscape photography) and C (automotive photography) do not have the same interests.
>
> 3. With the edge set, we learn to assign the adaptive distance-aware weights to all edges in $\mathcal{E}_e$. We believe that the edge weight between A and D is less than the edge weight between A and B. With the shared interest (landscape photography) between A and B, the distance between A and B is smaller than that between A and D. Furthermore, the edge weight between A and D is less than the edge weight between A and B (Proposition 2).
>
> Afterward, AdE generates a subgraph for the hyperedge $e$, with edges $\mathcal{E}_e$ = {(B, D), (B, A), (B, C), (D, A), (D, C)} and corresponding adaptive-aware weights. We hope our answer addresses your concern.
>
> *$\textbf{Q5:}$ How does AdE work on a hypergraph where nodes do not have attributes?*
>
> $\textbf{A5:}$ Our proposed method AdE is a hypergraph expansion that transforms a hypergraph into a graph. Our mechanisms, GSi-Net and the kernel function for edge weights, take the attribute features into account. Therefore, our proposed method can not be applied to hypergraphs without node attribute features directly.
>
> However, some techniques can be leveraged to generate node attribute features, such as identity features (one-hot encoded vector), constant features (constant value for each vector), and positional encoding (attribute based on the position of the node in the graph/hypergraph). With these generated attribute features, our proposed method still works. If there are some extreme scenarios in which attribute features are impossible to obtain, we think this could be our future work to expand the scalability of AdE.

---

> ### Author Response · Authors · 2023-11-17
> **Response to Reviewer 1ZMR (5/5)**
>
> *$\textbf{Q6:}$ Provide more details on the hypergraph datasets, such as their size, average degree distribution, average hyperedge size, etc.*
>
>
> $\textbf{A6:}$ Thank you for your suggestions, but we already provided a detailed description of all statistics of five benchmark hypergraph datasets in Table 3 of the Appendix in our original manuscript.  To make it clearer and more convenient, we also list the statistics of these datasets in the following Table 1 for your reference:
>
> ### Table 1. The statistics of benchmark hypergraph datasets for node classification tasks.
>
> |                       | Cora-CA | DBLP   | Cora  | Citeseer | Pubmed |
> |-----------------------|---------|--------|-------|----------|--------|
> | # nodes, $N$          | 2,708   | 41,302 | 2,708 | 3,312    | 19,717 |
> | # hyperedges, $M$     | 1,072   | 22,363 | 1,579 | 1,079    | 7,963  |
> | # features            | 1,433   | 1,425  | 1,433 | 3,703    | 500    |
> | # class               | 7       | 6      | 7     | 6        | 3      |
> | avg. $d(e)$           | 4.28    | 4.45   | 3.03  | 3.20     | 4.35   |
> | avg. $d(v)$           | 1.69    | 2.41   | 1.77  | 1.04     | 1.76   |
>
> \
> *$\textbf{Q7:}$ Does the proposed method apply to tasks other than node classification? (such as clustering, hyperedge prediction, etc.)*
>
> $\textbf{A7:}$ Thank you for your question. Yes, our proposed method AdE can be applied to tasks other than node classification, i.e.,  community detection, clustering, and link (edge) prediction. With the weighted graph generated by AdE, we employ graph neural networks to generate embeddings and further leverage the learned embeddings for the various downstream tasks, e.g., community detection, clustering algorithms, and link prediction.
>
> Here, Table 2 lists the results of CEGCN, HyperGCN, and AdE, over Cora, Cora-CA, and Citeseer datasets for hyperedge prediction tasks.
> We follow the work NHP [10] to conduct hyperedge prediction tasks. Specifically, after we generate a weighted graph via AdE and obtain learned embeddings from graph neural networks, we employ a **hyperlink scoring layer** from NHP:
>
>  $I_e =\sigma(W\cdot g(\\{h_v\\}\_{v\in e})+b), \\ \\ \\  \\
>  \\ \\  \\ (4)
> $
>
> where $W\in\mathbb{R}^{1\times d}$ is a learnable weight matrix, $g(\cdot)$ is a pooling function, e.g., mean pooling and sum pooling. As mentioned by NHP, the score $I_e$ for hyperedge $e$ needs to be higher than that for any set of nodes that does not form a hyperedge in the hypergraph. Therefore, the objective function is formulated as follows:
>
> $\mathcal{L} = \frac{1}{|\mathcal{E}|}\sum\_{e\in\mathcal{E}}\Lambda((\frac{1}{|\mathcal{F}|}\sum\_{f\in\mathcal{F}}I\_{f}) - I_e), \\ \\ \\  \\
>  \\ \\  \\ (5)
> $
>
> where $\mathcal{F}$ is a set of sampled hyperedges, and $I_f$ is the score of the sampled hyperedge $f\in\mathcal{F}$. Here, $\Lambda(\cdot)$ is the logistic function, i.e., $\Lambda(x)=\log(1+e^x)$.  The loss $\mathcal{L}$ tries to maximize the number of hyperedge scores (in $\mathcal{E}$) that are higher than the average score of the sampled hyperedges in $\mathcal{F}$. A more detailed discussion is provided in NHP [10]. For experiments, we follow the same settings in NHP, i.e., sample a set of $|\mathcal{E}|$ hyperedges, denoted as $\mathcal{F}$, such that the average degree of $\mathcal{F}$ approximately equals to the average degree of
> $\mathcal{E}$, and each newly generated hyperedge does not overlap with the original hyperedge set $\mathcal{E}$. Afterward, we concatenate hyperedge sets $\mathcal{F}$ and $\mathcal{E}$ to obtain a new hypergraph $\mathcal{G}'=(\mathcal{V}, \mathcal{E}',\mathcal{X})$. The hyperedges $\mathcal{E}'$ are randomly divided into training, validation, and test sets, with ratios of 50%, 25%, and 25%, respectively. We conduct experiments on two pooling functions, i.e., mean and sum. According to Table 2, our method AdE outperforms CEGCN and HyperGCN in all three datasets.
>
> ### Table 2. Performance comparison (Mean accuracy \%±  std ) of CEGCN, HyperGCN, and AdE over Cora, Cora-CA, and Citesser datasets for hyperedge prediction tasks.
>
> | Model | Pooling |Cora | Cora-CA| Citeseer |
> |---|---|---|---|---|
> |CEGCN| Mean | 70.38 ± 2.87|53.54 ± 2.06|74.50 ± 1.48|
> | | Sum | 79.27 ± 0.36 |  60.38 ± 2.87 | 74.50 ± 1.48|
> |HyperGCN| Mean | 81.16 ± 1.51|55.51 ± 3.14|77.24 ± 1.92|
> | | Sum | 81.01 ± 0.99 |61.32 ± 2.71 |  76.51 ± 2.82|
> |AdE| Mean |84.33 ± 0.99|56.09 ± 1.57|$\textbf{78.73 ± 1.98}$|
> | | Sum | $\textbf{85.53 ± 0.81}$| $\textbf{ 61.47 ± 1.63}$  | 77.97 ± 2.09|
>
> \
> Due to limited time during rebuttal, we are not able to conduct comprehensive experiments for link prediction tasks. However, we will update the comprehensive experiments in a later revision.
>
> We appreciate the opportunity to address your concerns and hope our response has provided satisfactory explanations. Please do not hesitate to let us know if you have any further comments or questions.
>
> [10] NHP: Neural Hypergraph Link Prediction. CIKM 2020.

---

> > ### Comment · Reviewer_1ZMR · 2023-11-17
> > **Thank you for the response.**
> >
> > Thank you for the response. Most of my concerns are answered. A suggestion is to update the manuscript to reflect these answers (left to the authors for better judgment). I have revised my rating.

---

> > > ### Author Response · Authors · 2023-11-17
> > > **Thank you for raising your rating.**
> > >
> > > Thank you for raising your rating. We appreciate your constructive comments and suggestions. We already updated the manuscript and most of the answers are highlighted in blue in the revision: A1 (Appendix H), A2 (Table 3 in Appendix), A3 (Section 4.1), A4 (Appendix B), A6 (Table 3 in Appendix), A7 (Appendix G). Please feel free to let us know if you have any further questions!

---

### Official Review · Reviewer_UEpe · 2023-11-02

**Soundness:** 3 good
**Presentation:** 3 good
**Contribution:** 3 good
**Rating:** 8
**Confidence:** 5

**Summary:**

This work proposes an adaptive method called AdE to convert the hypergraph into a weighted graph in an adaptive manner. First, it designs a global simulation network called GSi-Net to select two representative nodes to represent the corresponding hyperedge adaptively. After connecting the rest of the nodes with the two representative nodes, this work designs a distance-aware kernel function to dynamically learn the edge weights among node pairs. With the adaptive weighted graph, AdE leverages GNNs to learn the graph representatives for the downstream classification tasks. Moreover, this work provides comprehensive experiments and extensive theoretical justifications to demonstrate the effectiveness and rationality of AdE.

**Strengths:**

1. The idea of adaptively learning a weighted graph from the hypergraph is useful. Most existing works merely expand the hypergraph into a graph roughly and feed the converted graph into neural networks for representation learning, which results in information loss or information redundancy.
2. From my perspective, the model design of AdT including GSi-Net and the distance-aware kernel function is novel and rational, with theoretical proof and extensive experiments over five benchmark datasets.
3. Comprehensive experiments including comparison experiments, ablation studies, embedding visualization, and complexity analysis are conducted to demonstrate the effectiveness and its generalization.
4. The presentation is clear, making it easy to follow the key points of this work.

**Weaknesses:**

1. This work discusses three existing expansion methods, i.e., clique expansion, line expansion, and star expansion.  As this work introduces these three methods briefly in related Works, I do not get the difference between AdE, CE, LE, and SE. I am curious about the advantages and disadvantages of each method.
2. This work claims that AdE is equivalent to the weighted clique expansion in 3-uniform hypergraphs in Proposition 4. I am a bit confused about that as AdE is designed to dynamically learn the edge weights among node pairs. But most existing works, let us say, weighted clique expansion, assign a fixed weight between the node pair. Why is AdE equal to weighted clique expansion from the edge weight perspective?
3. This work conducts experiments over five benchmark hypergraph datasets, i.e., Cora-CA, DBLP, Cora, Citeseer, and Pubmed. I am a bit curious about the format of these datasets. It looks like these datasets are commonly used in graphs.  What are the differences between these hypergraphs and graphs? Do hyperedges in these hypergraphs have hyperedge features?

**Questions:**

According to the weakness, I list my questions as follows:
1. What are the advantages of AdE compared with these expansion methods?
2. In Proposition 4, why AdE equal the weighted clique expansion from the edge weight perspective?
3. What are the differences between these benchmark hypergraphs and the corresponding graphs?
4. Does the hyperedge in hypergraphs have the attribute features? Does the node in hypergraphs share the same attribute features as graphs?

**Details Of Ethics Concerns:**

NA.

---

> ### Author Response · Authors · 2023-11-15
> **Response to Reviewer UEpe (1/3)**
>
> We appreciate your valuable comments and would like to respond to your questions and concerns as follows.
>
> *$\textbf{Q1:}$ Discussion about existing hypergraph expansions.*
>
> $\textbf{A1:}$ Thank you for your comment. We would like to explain the difference between clique expansion (CE), line expansion (LE), and star expansion (SE), and further discuss the pros and cons of each method. We will discuss our proposed method AdE in $\textbf{A2}$.
>
> For a better understanding of hypergraph expansion methods, we will explain each method with Figure 1 of our paper, which illustrates classic hypergraph expansion methods, including CE, SE, and LE.
>
> Clique expansion (from hypergraph $\mathcal{H}$ to graph $\mathcal{G}_c$ in Figure 1) generates edges for every node pair within the hyperedge. Taking hypergraph $\mathcal{H}$ as an example, hyperedge $e_1$ has three nodes $v_1$, $v_2$, and $v_3$, so edges $(v_1, v_2), (v_1, v_3)$, and $(v_2, v_3)$ are added to the graph $\mathcal{G}_c$, which forms the "triangle shape" in the left of graph $\mathcal{G}_c$. For hyperedge $e_2$, since there are only two nodes, the edge of those two nodes is added into $\mathcal{G}_c$. The hyperedge $e_3$ is similar to hyperedge $e_1$ and forms the "triangle shape" in the right of graph $\mathcal{G}_c$.
>
> Star expansion (from hypergraph $\mathcal{H}$ to graph $\mathcal{G}_s$ in Figure 1) introduces a set of nodes that represents hyperedges, i.e., node $e_1$, $e_2$, and $e_3$ in $\mathcal{G}_s$. Then, each node is connected to the hyperedge node if it belongs to the corresponding hyperedge. For example, node $v_1$ belongs to hyperedge $e_1$ in hypergraph $\mathcal{H}$, therefore, node $v_1$ is connected to node $e_1$ in graph $\mathcal{G}_s$. Similarly, node $v_3$ is in hyperedges $e_1$ and $e_2$ in hypergraph $\mathcal{H}$,  therefore, node $v_1$ is connected to node $e_1$ and node $e_2$, respectively, in graph $\mathcal{G}_s$.
>
> Line expansion (from hypergraph $\mathcal{H}$ to graph $\mathcal{G}_l$ in Figure 1) creates a node in graph $\mathcal{G}_l$ for each node-hyperedge pair in hypergraph $\mathcal{H}$. For instance, node $v_6$ belongs to hyperedge $e_3$ in hypergraph $\mathcal{H}$,  so line expansion adds a node $v\_6\^{e_3}$ in to the graph  $\mathcal{G}_l$. For graph structure in $\mathcal{G}_l$, a node pair is connected if the node pair shares the same node or hyperedge in hypergraph  $\mathcal{H}$. For example, $v\_6\^{e_3}$ connects to $v\_4\^{e_3}$, because those two nodes are both belongs to hyperedge $e_3$.
>
> Next, we will summarize the difference between CE, SE, and LE with their pros and cons:
>
> 1. SE and LE introduce new nodes to represent the hyperedges and node-hyperedge pairs, respectively, while our method AdE and CE use the same nodes in the original hypergraph.
>
> 2. In the star expansion approach, an edge connects node $v_i$ to hyperedge node $e_i$, eschewing direct node-to-node connections. This approach inherently facilitates edge-related downstream tasks by leveraging the attribute features of hyperedge nodes. However, it treats the hypergraph as a bipartite graph and introduces a heterogeneity between nodes and hyperedge nodes. Such heterogeneity potentially raises information loss and noise within the graph structure. Moreover, the absence of direct node-to-node connections in the graph precludes the message-passing function from directly passing information between nodes. Besides, the expanded graph is too complicated for existing well-studied graph algorithms, mostly designed for simple graphs.
>
> 3. Line expansion transforms a hypergraph into a homogeneous graph, introducing new nodes in the expanded graphs. Unlike star expansion, line expansion preserves homogeneity, as all nodes in the graph represent pairs of "node-hyperedge" from the original hypergraph. However, this expansion leads to a substantially large graph, significantly increasing the computational complexity and space complexity.
>
> 4. Clique expansion is a relatively straightforward expansion method, and it supports all graph-based algorithms. However, it introduces noise or information redundancy as it connects every node pair in a hyperedge.
>
> We hope our explanation answers your question.

---

> ### Author Response · Authors · 2023-11-15
> **Response to Reviewer UEpe (2/3)**
>
> *$\textbf{Q2:}$ What are the advantages of AdE compared with these expansion methods?*
>
> $\textbf{A2:}$ The advantages of AdE can be summarized as follows:
>
> 1. Unlike other classic hypergraph expansions discussed in $\textbf{A1}$ that merely construct a graph based on the hypergraph structure, AdE considers node attribute features in the construction process. Specifically, AdE leverages GSi-Net to learn the importance of each feature dimension and further generates scaled attribute features to select two representative nodes for representing the information for each hyperedge. With GSi-Net, our model AdE learns to find the most representative node pair for the corresponding hyperedge. Simply selecting two representative nodes during hypergraph expansion, such as HyperGCN is unable to find the most representative node for each hyperedge.
>
> 2. AdE also introduces a novel distance-aware kernel function, i.e., $\mathcal{W}\_{i,j} =\exp(-\frac{1}{b}\sum\_{d=1}^b\frac{ \mathcal{U}\_{i,j}(\mathbf{X}\_{\text{a},({i,d})} - \mathbf{X}\_{\text{a}, (j, d)})^2}{\theta\_d^2}) $, to adaptively learn the edge weights to ensure that the edge between two nodes with similar attribute features will have a higher weight during hypergraph conversion.
> In contrast to AdE, classic hypergraph expansion methods generate edges with fixed weights. Specifically, CE, SE, and LE assign a fixed edge weight (e.g., 1) to each edge generated during hypergraph conversion. In HyperGCN, for each hyperedge $e$, the hypergraph expansion creates edges with a fixed weight of $\frac{1}{2|e|-3}$. Proposition 3 theoretically justifies that our novel distance-aware kernel function $\mathcal{W}$ enhances HyperGCN by generating more adaptive weighted graphs. To conclude, the distance-aware kernel function in AdE can learn more accurate and rational edge weights than existing methods, which is more applicable in real-world scenarios.
>
> Next, we would like to intuitively show the advantages of AdE with an example. Suppose we have a hyperedge $e$  that four users (A, B, C, and D) have interests in photographs.  Node A is interested in landscape photography, Node B enjoys automotive and landscape photography, Node C is interested in automotive photography, and Node D likes food photography. Based on the above scenario, classic methods, i.e., clique expansion, will generate an edge set $\mathcal{E}_e$ ={(A,B), (A,C), (A,D), (B,C), (B,D), (C,D)}. However, Nodes A and C do not share any interests and they are supposed to be disconnected or connected with very small edge weights. But our AdE can professionally handle real-world scenarios:
>
> 1. AdE first selects two representative nodes with the largest distance within the hyperedge. Let us say that nodes B and D are selected as the representative nodes as their styles are very dissimilar, meaning that they have the longest distance in the attribute space.
>
> 2. We connect the rest of the nodes (i.e., A and C) with the two representative nodes, respectively, and we further obtain the edge set $\mathcal{E}_e$ = {(B, D), (B, A), (B, C), (D, A), (D, C)}.  Nodes A and C are not connected in $\mathcal{E}_e$, as A (landscape photography) and C (automotive photography) do not have the same interests.
>
> 3. With the edge set, we learn to assign the adaptive distance-aware weights to all edges in $\mathcal{E}_e$. We believe that the edge weight between A and D is less than the edge weight between A and B. With the shared interest (landscape photography) between A and B, the distance between A and B is smaller than that between A and D. So, the edge weight between A and D is smaller as our distance-aware kernel function tends to assign smaller weights for dissimilar node pairs.
>
> Therefore, AdE generates a subgraph for the hyperedge $e$, with edges $\mathcal{E}_e$ = {(B, D), (B, A), (B, C), (D, A), (D, C)} and corresponding adaptive-aware weights. We hope our detailed analysis and intuitive examples can address your concern.
>
> *$\textbf{Q3:}$ Why is AdE equivalent to the weight clique expansion in 3-uniform hypergraphs from the edge weight perspective?*
>
> $\textbf{A3:}$ In the justification of Proposition 4, the adjacency matrix of a weighted graph $\mathcal{G}_c$ can be formulated as follows (Equation 6 in the paper): $A\_{\text{c},(i, j)} = \sum\_{e\in\mathcal{E}} \mathbf{H}\_{i,e}\mathbf{H}\_{j,e} w\_{i,j}\^{(e)}$, where $w\_{i,j}\^{(e)}$ represents the weight of edge $(v\_i, v\_j)$ with the default value of 1, as discussed in $\textbf{A2}$. Our justification is based on the assumption that the edge weight $w\_{i,j}\^{(e)}$ in the previous equation is the normalized adaptive edge weight $\bar{\mathcal{W}}\_{i,j}^{(e)}$, computed in Equation 4 in the paper. Under this assumption, our proposed method, AdE, is equivalent to the weighted clique expansion in 3-uniform hypergraphs. Besides, we conduct experiments over AdE and weighted clique expansion to support our justifications.

---

> ### Author Response · Authors · 2023-11-15
> **Response to Reviewer UEpe (3/3)**
>
> *$\textbf{Q4:}$ What is the difference between these hypergraphs and graphs? Do hyperedges in these hypergraphs have hyperedge features? Does the node in hypergraphs share the same attribute features as graphs?*
>
> $\textbf{A4:}$ Thank you for the question. We would like to mention that all benchmark hypergraphs and the corresponding graphs share the same original information, but they still have some differences:
>
> 1. Hypergraphs try to extract the high-order group-wise relationships while graphs try to extract the pair-wise relationships. Let us take an example of the Cora-CA dataset in Table 1. Cora-CA hypergraph aims to depict the group-wise relationships that all of the papers that the author contributes to, showing his research interests in an easier way. However, Cora-CA graph aims to extract the pair-wise relationships that these two authors contribute to the same paper, showing its individual relationships in graphs.
>
> 2. They are designed for different downstream tasks. Let us also take Cora-CA in Table 1 as an example. The Cora-CA hypergraph is designed for the research fields of the paper, while the Cora-CA graph aims to classify the research interests of the authors.
>
> We also list the differences of all datasets in Table 1 for your reference. Hopefully, our response can address your first concern.
>
>
> ### Table 1. Hypergraph and graph dataset comparison.
>
> | Network Type| Element Type | Cora-CA | DBLP | Cora | Citeseer | Pubmed |
> |----------|----------|----------|----------|----------|----------|----------|
> | Hypergraph| Node | Paper   | Paper | Paper | Paper | Paper |
> | Hypergraph| Hyperedge | Author| Author  | Citation | Citation  | Citation  |
> | Graph| Node | Author |Author | Paper | Paper | Paper |
> | Graph| Edge  | Co-author| Co-author | Citation | Citation | Citation  |
>
> For the second question, all of these hypergraph datasets have node attribute features, while none of these datasets have the hyperedge attribute features. Besides, these nodes in hypergraphs share the same node attribute features as those in the corresponding graph datasets.
>
> We hope that our response has addressed your concerns and that you acknowledge the contribution of our novel model design. Please let us know if you have any additional comments.

---

> > ### Comment · Reviewer_UEpe · 2023-11-19
> > **Appreciate the authors' responses.**
> >
> > I appreciate the authors' responses, which have addressed my concerns and questions. Consequently, I would like to raise my score to indicate support.

---

> ### Author Response · Authors · 2023-11-20
> **Thank you for rasing your rating**
>
> Thank you for raising your rating to endorse our work. We appreciate your acknowledgment of the novelty and contribution of this work. Please also feel free if you have any further comments or questions.

---

### Official Review · Reviewer_qSDB · 2023-11-03

**Soundness:** 2 fair
**Presentation:** 2 fair
**Contribution:** 2 fair
**Rating:** 3
**Confidence:** 5

**Summary:**

This paper proposes a new hypergraph learning method based on the previous work of HyeprGCN, by expanding the hypergraph into a weighted graph and then running a graph neural network over the expanded graph. Experiments show that the proposed method marginally outperforms selected baselines.

**Strengths:**

1. The authors have made a good observation that hypergraph expansion has grown to be an important area of study in hypergraph-related learning and analysis. Therefore, I think the subject of study is very meaningful.

2. The experiments are well designed to support the main claims in the methodology. They are also quite extensive in coverage of various expansion methods (Table 1) as well as hypergraph learning methods (Table 2)

**Weaknesses:**

1. I am not fully convinced about the design choice that, regardless of the size of the hyperedge, always two nodes are chosen as representative nodes. Why is it not three or an adaptive number of representative nodes? Also, in expansion, this essentially always approximate any internal connectivity within a hypergraph by a "bar-bell" shaped graph, which does not look very intuitive to me. Can the authors explain more on this matter?

2. I remains very unclear to me why we should use S, the sum of scaled attribute features, to select representative nodes. In other words, why would having a large (or small) sum of scaled attributes be an important factor to consider when choosing the representative nodes?

3. The propositions 1-4 all seem to discuss something straightforward to me. It is easy to see that the distance metric defined in Eq. (3) is non-negative, commutative, and assign higher values to more similar pairs of attributes due to its weighted sum of square.  For Proposition 4, AdE by its design would obviously connect every pair of nodes in a hyperedge with only 3 nodes.

4. The numerical results in both Table 1 and Table 2 show that the proposed method is very marginally higher than the best baseline.

Minor typo:
"... in a 3-unifirm hyperedge", in the last paragraph of Section 4;

**Questions:**

See Weaknesses.

---

> ### Author Response · Authors · 2023-11-15
> **Response to Reviewer qSDB (1/4)**
>
> We appreciate your valuable comments and would like to respond to your questions and concerns as follows.
>
> *$\textbf{Q1:}$ Why do we choose two nodes as representative nodes? Why is it not three or an adaptive number of representative nodes?*
>
> $\textbf{A1:}$ Thanks for your constructive comments. We would like to answer your questions and explain our AdE step by step. First of all, existing classic expansion methods struggle to preserve the connection information during expansion. For instance, clique expansion (CE) [1] connects every node pair within the hyperedges when converting a hypergraph to a graph, resulting in information redundancy.  In light of this, a recent work, HyperGCN [2], proposes a novel method called Hypergraph Laplacian with mediators to convert hypergraphs into weighted graphs. Specifically, for each hyperedge $e$, HyperGCN selects two nodes based on the following rule:
>
> $$
> (v\_{e^+}, v\_{e^-})=\arg\max\_{v_i,v_j\in e}|\mathbf{S}_i - \mathbf{S}_j|,  \\  \text{where}   \\  \mathbf{S}= \mathbf{X} \cdot W\_\text{r} . \\ \\ \\  \\
>  \\ \\  \\ (1)
> $$
>
> Here, $W\_\text{r}$ is a **random** feature weight and $\mathbf{S}\in\mathbb{R}^N$. HyperGCN is inspired by existing works [3, 4] that **two nodes with the largest distance within the hyperedge can represent the information within the hyperedge to a large extent**. Then, HyperGCN connects all other nodes with two representative nodes $v\_{e^+}$ and $v\_{e^-}$, respectively. Afterward, each edge in the weighted graph is assigned a fixed weight in HyperGCN. Please refer to Figure 2 in [2] for more details. However, HyperGCN still faces several challenges: (i) the selected two nodes are not representative enough as it is generated based on the random information $\mathbf{S} =\mathbf{X}\cdot W_r$.  Here $W_r$ is a random matrix.  (ii) The edge weight for each node pair is fixed as $\frac{1}{2|e|-3}$. It does not consider the variety of edge weights among node pairs.
>
> In light of this, we first design a GSi-Net module to select the two representative nodes $(v\_{e^+}, v\_{e^-})$ adaptively via function:
>
> $$(v\_{e^+}, v\_{e^-})=\arg\max\_{v_i,v_j\in e}|\mathbf{S}_i - \mathbf{S}_j|, \text{where} \\ \\ \mathbf{S}=\sum\_{k=1}^b\mathbf{X}\_{:,k}\cdot
> \sigma(W_2\cdot\text{ReLU}(W_1\cdot\textbf{X}\_\text{g})).
>   \\ \\ \\  \\  \\ \\  \\ (2) $$
>
> Here,  $\mathbf{S}\in\mathbb{R}^N$, and $\textbf{X}_\text{g}$ is the global level representation of attribute features in GSi-Net. Compared to Eq. 2 in GSi-Net and Eq.1 in HyperGCN, we believe our real-value signal $\mathbf{S}$ (in Eq. 2) learned by GSi-Net is more informative and sufficient to select representative nodes for hyperedges. To handle the second challenge, we design a distance-aware kernel function to adaptively assign the edge weights $\mathcal{W}\_{i,j}$ to edge $(v_i,v_j)$, which is formulated as follows:
>
> $$\mathcal{W}\_{i,j} =\exp(-\frac{1}{b}\sum\_{d=1}^b\frac{ \mathcal{U}\_{i,j}(\mathbf{X}\_{\text{a},({i,d})} - \mathbf{X}\_{\text{a}, (j, d)})^2}{\theta\_d^2}). \\ \\ \\  \\
>  \\ \\  \\ (3) $$
>
> Compared Eq. 3 to the fixed weight $\frac{1}{2|e|-3}$ in HyperGCN, our adaptive edge weight tends to assign higher edge weights between similar node pairs and lower edge weights between dissimilar node pairs. In order to compare AdE with existing work HyperGCN intuitively, we add a new figure to the revised manuscript. Please refer to Figure 5 for clarification. We hope the above clarification about AdE can facilitate your understanding.
>
> [1] Hypergraph spectral learning for multi-label classification, KDD 2008.
>
> [2] HyperGCN: A New Method of Training Graph Convolutional Networks on Hypergraphs, NeuIPS 2019.

---

> ### Author Response · Authors · 2023-11-15
> **Response to Reviewer qSDB (2/4)**
>
> *$\textbf{Q2:}$ Why do we expand a hypergraph into a "bar-bell" shaped graph?*
>
> $\textbf{A2:}$ Thank you for the question, but we are unclear about your questions. According to [3,4], bar-bell graphs refer to the graph obtained by connecting n-copies of a complete graph by a bridge. We do not think our weighted graphs are bar-bell graphs. But it may cause some confusion that the shape looks like bar-bell graphs. Sorry for the confusion. We would like to explain how AdE converts hypergraphs into weighted graphs as follows:
> 1. For each hyperedge $e\in\mathcal{E}$, AdE selects two representative nodes according to the rule: $(v\_{e^+}, v\_{e^-})=\arg\max\_{v_i,v_j\in e}|\mathbf{S}_i - \mathbf{S}_j|$, where $\mathbf{S}=\sum\_{k=1}^b\mathbf{X}\_{:,k}\cdot
> \sigma(W_2\cdot\text{ReLU}(W_1\cdot\textbf{X}\_\text{g}))$ . The rest of nodes in hyperedge $e$, i.e., $\mathcal{V}\_m^e=\\{v_m|v_m\ne v\_{e^+}, v_m\ne v\_{e^-}, v_m\in e\\}$ are mediators.
> 2. For each hyperedge $e\in\mathcal{E}$, we connect each mediator with two representative nodes $v\_{e^+}$ and $v\_{e^-}$, respectively, and further obtain the edge set $\mathcal{E}_e =\\{  \\{v\_{e^+}, v\_{e^-}\\},\\{v\_{e^+}, v\_m\\}, \\{v\_{e^-}, v\_m\\}| v_m\in \mathcal{V}_m^e \\}$.
> 3. We compute edge weight in each edge set  $\mathcal{E}_e$ via our designed learnable kernel function $\mathcal{W}\_{i,j}$ (Eq. 3), and normalize these edge weights with respect to hyperedges, $\bar{\mathcal{W}}\_{i,j}^{(e)} = \frac{\mathcal{W}\_{i,j}}{\sum\_{\\{v_k,v_j\\}\in\mathcal{E}_e}\mathcal{W}\_{k,g}}$.
>
> By the aforementioned steps, we obtain the weighted graph with the adaptive adjacency matrix $A_\text{a}=\sum\_{e\in\mathcal{E}}\mathbb{I}[\{v_i, v_j\}\in\mathcal{E}_e]  \bar{\mathcal{W}}\_{i,j}^{(e)}$.
>
>
> [3] Computing Kemeny's constant for a barbell graph. The Electronic Journal of Linear Algebra 35 (2019): 583-598.
>
> [4] On the k-Metric Dimension of a Barbell Graph and a t-fold Wheel Graph. Mathematical Combinatorics 4 (2018): 146-159.

---

> ### Author Response · Authors · 2023-11-15
> **Response to Reviewer qSDB (3/4)**
>
> *$\textbf{Q3:}$ Why would having a large (or small) sum of scaled attributes be an important factor to consider when choosing the representative nodes?*
>
> $\textbf{A3:}$ We understand that leveraging $\mathbf{S}$ as criteria to choose representative nodes may confuse, and we would like to explain it in detail. According to $\mathbf{A1}$, as stated in previous works [5, 6], **the node pair with the largest distance in the attribute feature space can represent the hyperedge information to a large extent.**  According to Eq. 1 and Eq. 2 in $\mathbf{A1}$, both HyperGCN and AdE propose to select two representative nodes for symbolizing the content within corresponding hyperedges based on the attribute features.  Here, in AdE, we leverage the sum of the scaled attribute feature to improve the **efficiency** during hypergraph expansion. Suppose we leverage the most straightforward way that $\mathbf{S}= \mathbf{X}$, $\mathbf{S}\in\mathbb{R}^{N*b}$, where $N$ is the number of nodes and $b$ is the dimension of attribute feature,  and we calculate the distance (e.g., Euclidean distance) for finding the node pair with the largest distance in each hyperedge, the time complexity could be in **quadratic time** in the worst case. If we employ the sum of scaled attribute feature where $\mathbf{S}=\sum\_{k=1}^b\mathbf{X}\_{:,k}\cdot \sigma(W_2\cdot\text{ReLU}(W_1\cdot\textbf{X}\_\text{g}))$, here $\mathbf{S}\in\mathbb{R}^N$, the time complexity is $\mathcal{O}(N)$, which is **linear** to the node size $N$. HyperGCN uses the similar criteria $\mathbf{S} = \mathbf{X}\cdot W\_\text{r}$ with random feature weight $W\_\text{r}$ and $\mathbf{S}\in\mathbb{R}^N$ to select the representative node pair. I hope the explanations can address your questions.
>
>
> *$\textbf{Q4:}$ Discussion about the proposition.*
>
> $\textbf{A4:}$ We appreciate your feedback on the propositions. Our intent with these propositions is to provide a clear and rigorous analysis supporting our model design, even if some aspects might seem straightforward. In Proposition 1 and 2, we aim to demonstrate that with the prior information $\mathcal{U}$, our function remains a distance-aware kernel function. For Proposition 3, it is necessary to justify that our proposed method generates more adaptive weighted graphs than HyperGCN since our approach draws inspiration from it. For Proposition 4, this part of the justification is based on the findings of the ablation study. We observe that for the benchmark dataset Cora and Citeseer, the performance of CE with our adaptive kernel function is similar to that of the AdE model without GSi-Net. This finding inspired us to analyze the relationship between clique expansion and our proposed AdE method. Proposition 4 theoretically justifies our observations. Although these propositions might seem straightforward, we believe they are necessary to support our model design. We hope our response answers your concern.
>
>
> [5] Re-revisiting learning on hypergraphs: Confidence interval and subgradient method. ICML 2017.
>
> [6] Learning with hypergraphs: Clustering, classification, and embedding. NeurIPS 2016.

---

> ### Author Response · Authors · 2023-11-15
> **Response to Reviewer qSDB (4/4)**
>
> *$\textbf{Q5:}$ The performance of the model is not significant.*
>
> $\textbf{A5:}$  We appreciate your observation that the performance of some datasets does not outperform the best baseline very much. We believe even small improvements are significant in this domain. Here, we take two popular methods as examples. Table 1 provides the accuracy performance and the corresponding improvements compared with **AllSets**[7] and the best baseline over 15 datasets. Table 1 highlights the improvements that are less than 1.  The improvements in **eleven out of fifteen datasets** are less than 1, and the average improvement over fifteen datasets is 0.425.
>
> ### Table 1 The best result of mean accuracy in Allset.
>
> | | Cora | Citeseer| Pubmed | Cora-CA | DBLP-CA| Zoo| 20Newsgroups| mushroom|
> |---|---|---|---|---|---|---|---|---|
> |Allset|78.59|73.08|88.75|83.63|91.53|97.50|81.38|100.00|
> |Best Baseline|80.18|74.12|88.25|84.04|91.69|93.65|81.42|100.00|
> |Improvement|$\textbf{-1.59}$|$\textbf{-1.04}$|$\textbf{0.50}$|$\textbf{-0.41}$|$\textbf{-0.16}$|3.85|$\textbf{-0.04}$|0.00|
>
> | |NTU2012|ModelNet40|Yelp|House(1)|Walmart(1)|House(0.6)|Walmart(0.6)|
> |---|---|---|---|---|---|---|---|
> |Allset|88.69|98.20|36.89|69.33|65.46|83.14|78.46|
> |Best Baseline|89.30|98.07|33.04|71.05|62.45|83.27|77.72
> |Improvement|$\textbf{-0.61}$|$\textbf{0.13}$|3.85|$\textbf{-1.72}$|3.01|$\textbf{-0.13}$|$\textbf{0.74}$|
>
>
> Table 2 shows the best results of the mean test error (lower is better) in **HyperGCN**[2]. According to Table 2, the improvement in Cora and Citeseer is only 0.04 and 0.05, respectively and the average improvement of HyperGCN over five datasets is 1.464.
>
> ### Table 2. The best results of the mean test error (lower is better) in HyperGCN.
>
> |   | DBLP | Pubmed | Cora-CA | Cora| Citeseer |
> |----------|----------|----------|----------|----------|----------|
> | Best Baseline |25.65|29.41|31.90|32.41|37.40|
> |Best HyperGCN variant|24.09|25.56|30.08|32.37|37.35|
> |Improvement|1.56 | 3.85 | 1.82| $\textbf{0.04}$| $\textbf{0.05}$|
>
> From the above two tables, we can find even the state-of-the-art methods (e.g., AllSet and HyperGCN) are not satisfactory enough over all datasets, and the relatively high improvement is not always guaranteed over all datasets. The main contribution of this work is that we propose a novel and adaptive expansion method to convert hypergraphs into graphs.
>
>
> [7] You are AllSet: A Multiset Function Framework for Hypergraph Neural Networks, ICLR 2022.
>
> Thank you for pointing out the typo. We already corrected them in the revised manuscript. We appreciate the opportunity to address your concerns and hope our response has provided satisfactory explanations. We realize that most of the concerns are regarding section 4.1 in our paper. So we revised our manuscript and highlighted the changes in blue. We would like to emphasize the contribution of our novel model design and sincerely hope that you recognize its value. Please do not hesitate to let us know if you have any further comments or questions.

---

> > ### Comment · Reviewer_qSDB · 2023-11-19
> > **Thanks for the explanation**
> >
> > I appreciate the explanation and extra experiments by authors. However, a few prominent concerns of mine still exist. It would be great if the authors can provide further explanations:
> >
> > 1. I am still not convinced that the experiment result is significant, based on the simple statistical fact that on many datasets, the improvement over the best baselines is within one standard deviation. The fact that Allset or HyperGCN do not outperform baselines very much does not justify the significance of this method.
> >
> > 2. In A1, the authors refer to [3, 4] to explain why always 2 nodes are always used as the backbone in expansion. However, [3,4] have not explained this at all, so I am still very confused on this matter. Can you please provide more principled insight why the proposed, specific type of expansion, which uses 2 nodes as backbone, is good for preserving hyperedge information? It would also be a very satisfying answer if you could please show me the reason of  why not selecting 3 nodes or, in particular, an adaptive number of nodes?
> >
> > 3. I can understand the main contribution of this work is that it refines HyperGCN by (1) refining the node selection process by adding a scaling mask to the attributes, and (2) replacing its uniform projection by weighted projection. I am not sure if this contribution looks a bit incremental, especially considering that its performance does not have much advantage over the baselines.
> >
> > 4. I still do not think the propositions are appropriate enough as something that requires a formal proof, especially when their proof are also put in the main text. This is mostly a writing issue though, just that I do not think they help much with establishing technical depth.

---

> > > ### Author Response · Authors · 2023-11-20
> > > **Response to Follow-up Comments of Reviewer qSDB (3/3)**
> > >
> > > *$\textbf{Q4:}$ Propositions seem to be inappropriate enough as something that requires a formal proof, especially when their proof is also put in the main text. These propositions do not help much with establishing technical depth.*
> > >
> > > $\textbf{A4:}$ We appreciate your constructive comments. According to the introduction in Wikipedia, a proposition is a statement of lesser importance or one that is considered so elementary or immediately obvious, that it may be stated without proof, while a theorem is a statement that needs to be proven [5]. We think some propositions (Proposition 1, 2, and 3) in our work are important to facilitate understanding of our designed method. Thank you for your suggestions. We already moved some theoretical justifications (Proposition 4) into the Appendix of the revised manuscript.
> > > \
> > > \
> > > [5] Theorem, https://en.wikipedia.org/wiki/Theorem, 2018.
> > > \
> > > \
> > > We really appreciate your comments and suggestions and we hope that our responses successfully alleviate all your concerns. If so, we would kindly ask you to adjust your review score. Please also let us know if you have any additional comments.

---

> > > > ### Comment · Reviewer_qSDB · 2023-11-22
> > > >
> > > > Thanks for this second-round response. I have very carefully read it. While I appreciate the authors' effort, I would like to maintain my current score, for 4 reasons:
> > > >
> > > > 1. The significance of the proposed method's empirical performance still seriously lacks justification. My concern that "the improvement over the best baselines is within one standard deviation" has not been addressed at all in the response. The authors have gone great lengths to establish that "Allset or HyperGCN does not outperform baselines very much". I truly appreciate their effort. However, this does not help at all to justify the empirical significance of the proposed method.
> > > >
> > > > 2. I am still not convinced of the idea's novelty and technical depth. It has been clear that the proposed method still mostly follow the HyperGCN's backbone. That is, (1) selecting two representative nodes for each hyperedge, (2) doing expansion by connnecting the two representative nodes with every other nodes in the hyperedge, and (3) runing a GNN on the expansion. I do not think that current refinement of node selection process contributes enough novelty or technical depths that are suitable for publication at ICLR this time -- especially given that the performance improvement is so marginal according to Table 1 and 2.
> > > >
> > > > 3. My concern about "why always selecting two representative nodes" is partially resolved. However, I do believe that the presentation of this paper should be significantly refined to provide enough **principled** introduction of this idea. It just seems too natural that for large hyperedges using three or four representative nodes, instead of two, can be beneficial to performance. However, this potential was never discussed and experimented.
> > > >
> > > > 4. My concern that "propositions are too simple and they should not have long proof in the main text" is partially resolved, in the sense that I agree we can lower the bar of technical depths for "propositions". However, I still believe (1) their proofs are still too long in this latest version. I would suggest putting all of them to Appendix, and spare more space to explain 2 and 3 above; (2) the propositions are still too straightforward.
> > > >
> > > > Based on the discussion above, I do not think the paper in its current form is ready for publishing. I have therefore decided to keep my current score.

---

> > > > > ### Author Response · Authors · 2023-11-23
> > > > > **Response to Third-round Comments of Reviewer qSDB (3/3)**
> > > > >
> > > > > *$\textbf{Q4:}$ My concern that "propositions are too simple and they should not have long proof in the main text" is partially resolved, in the sense that I agree we can lower the bar of technical depths for "propositions". However, I still believe (1) their proofs are still too long in this latest version. I would suggest putting all of them in the Appendix, and spare more space to explain 2 and 3 above; (2) the propositions are still too straightforward.*
> > > > >
> > > > > $\textbf{A4:}$ Thank you for your suggestions regarding the writing behaviors. We already move half of our propositions (i.e., Proposition 3 and 4) to the Appendix and shorten the proof about Proposition 1 and 2 in the main text. We think the Propositon 1 and 2 can help understand the main idea of our designed model. We appreciate your constructive comments on this.
> > > > >
> > > > > \
> > > > > [1] HyperGCN: A New Method of Training Graph Convolutional Networks on Hypergraphs, NeuIPS 2019.
> > > > >
> > > > > [2] Generalizing the Hypergraph Laplacian via a Diffusion Process with Mediators. COCOON 2019.
> > > > >
> > > > > [3] Spectral Properties of Hypergraph Laplacian and Approximation Algorithms, J. ACM 2018.
> > > > >
> > > > > [4] Hypergraph Markov Operators, Eigenvalues and Approximation Algorithms, STOC 2015.
> > > > >
> > > > > \
> > > > > \
> > > > > We really appreciate your valuable efforts and constructive suggestions. We hope our third-round explanation can assist you in understanding our model design comprehensively. AdE is designed as an adaptive hypergraph expansion framework that the **weighted graph can be generated dynamically over different downstream tasks**, which is **the first work that can learn to preserve the high-order relationships within hyperedges adaptively**. Although AdE shows comparable ability compared with SOTA over several datasets, it **outperforms the backbone HyperGCN significantly**, showing the effectiveness of our AdE. We sincerely hope that our explanation can alleviate all your concerns. If so, your consideration of adjusting your rating will be greatly appreciated.

---

> > > ### Author Response · Authors · 2023-11-21
> > > **Kinder Reminder to Reviewer qSDB**
> > >
> > > Dear Reviewer qSDB,
> > >
> > > We appreciate your constructive comments. As the deadline for rebuttal is coming up, we kindly remind you to review our responses.  If our responses have addressed all your concerns, we would greatly appreciate your consideration in adjusting your rating to support this work. Should you have additional questions, please do not hesitate to let us know! Thank you again for your valuable feedback and efforts!

---

> > > ### Author Response · Authors · 2023-11-22
> > > **Urgent Reminder to Reviewer qSDB**
> > >
> > > Dear Reviewer qSDB,
> > >
> > > We understand you may be busy during the rebuttal. As the rebuttal deadline is nearing, we would really appreciate your valuable time if you could take a moment to review our second-round response. We sincerely hope that our second-round response can alleviate all your concerns you may have. If you have any further questions, please always feel free to let us know!

---

> ### Author Response · Authors · 2023-11-20
> **Response to Follow-up Comments of Reviewer qSDB (1/3)**
>
> Thank you for your follow-up comments and we would like to answer all your questions as follows:
>
> *$\textbf{Q1:}$  The experiment result is not significant, based on the simple statistical fact that on many datasets. Besides, the improvement over the best baselines is within one standard deviation.*
>
> $\textbf{A1:}$ Thank you for your follow-up comments and we would like to demonstrate the effectiveness of our model design from two aspects:
>
> 1. From an **empirical** point of view, in our previous response A5, we provided the detailed performance of the state-of-the-art AllSet [1] (Table 1 in previous A5) and a comparable method HyperGCN [2] (Table 2 in previous A5) as we believe that comparing the improvement performance of AdE and the SOTA AllSet and HyperGCN can be convincing to show the effectiveness in the hypergraph learning field.
>
> To make the comparison clearer,  we also list the performance of AllSet in the following Table 2 for your convenience. From Table 2, we can find out that **SOTA AllSet fails to outperform the best baseline methods over eight out of fifteen (8/15) datasets (negative values in Table 1)**. Besides, the improvements in eleven out of fifteen datasets (11/15) are less than 1. However, we can not deny the contribution of AllSet which integrates DeepSets and hypergraph neural networks and further designs a novel hypergraph neural network, providing new insights and a succinct pipeline for future research fields in hypergraph representation learning.  Besides, we also list the performance of **HyperGCN** over six benchmark datasets (Table 3) and we find out that **three out of six datasets (1/2) have lower performance improvement of less than 0.1**.
> \
> Moreover, for a more obvious comparison to show the effectiveness of AdE in hypergraph learning, we further compare the average improvement of the SOTA AllSet, HyperGCN, and our model AdE in the following Table 1. From this table, **we find out the mean improvement of AdE in Table 1 is comparable and competitive** , which we believe can demonstrate our model AdE is competitive and effective enough in the hypergraph learning field.
>
> **Table 1**: **Mean performance improvement** comparison between AllSet, HyperGCN, and AdE. The mean performance improvement equals the average of the performance gap of the proposed method and the corresponding best baseline method over all datasets.
>
> |Method | Mean Improvement|
> |----|----|
> |AllSet|   0.42|
> |HyperGCN|  0.94 |
> |AdE| 0.73 |
> |||
>
> **Table 2**: Mean accuracy performance comparison between **AllSet** and the best baseline method over 15 datasets.
>
> | | Cora | Citeseer| Pubmed | Cora-CA | DBLP-CA| Zoo| 20Newsgroups| mushroom|
> |---|---|---|---|---|---|---|---|---|
> |AllSet|78.59|73.08|88.75|83.63|91.53|97.50|81.38|100.00|
> |Best Baseline|80.18|74.12|88.25|84.04|91.69|93.65|81.42|100.00|
> |Improvement|$\textbf{-1.59}$|$\textbf{-1.04}$|0.50|$\textbf{-0.41}$|$\textbf{-0.16}$|3.85|$\textbf{-0.04}$|0.00|
>
> | |NTU2012|ModelNet40|Yelp|House(1)|Walmart(1)|House(0.6)|Walmart(0.6)|
> |---|---|---|---|---|---|---|---|
> |Allset|88.69|98.20|36.89|69.33|65.46|83.14|78.46|
> |Best Baseline|89.30|98.07|33.04|71.05|62.45|83.27|77.72
> |Improvement|$\textbf{-0.61}$|0.13|3.85|$\textbf{-1.72}$|3.01|$\textbf{-0.13}$|0.74|
> |||||||||
>
> **Table 3**: Mean test error (lower is better) comparison between **HyperGCN** and the best baseline method over 6 datasets.
>
> |   | DBLP | Pubmed | Cora-CA | Cora| Citeseer | House |
> |----------|----------|----------|----------|----------|----------|----------|
> | Best Baseline |25.65|29.41|31.90|32.41|37.40| 66.37|
> |Best HyperGCN variant|24.09|25.56|30.08|32.37|37.35|64.71|
> |Improvement|1.56 | 3.85 | 1.82| 0.04| 0.05|$\textbf{-1.66}$|
> |||||||
>
>
> **Table 4**:  Accuracy comparison between **AdE** and the best baseline method over 7 datasets.
>
> |   | Cora-CA |DBLP | Cora | Citeseer| Pubmed| House1.0 | House0.6|
> |----------|----------|----------|----------|----------|----------|----------|----------|
> | Best Baseline |81.32|91.34|79.62|71.62|86.87|66.18| 80.74|
> |AdE|82.07|90.86|80.74|72.49|87.88|67.80|82.06|
> |Improvement|0.75| $\textbf{-0.48}$ | 1.12| 0.87| 1.01|1.62|1.32|
> ||||||||
>
> 2. From the **contribution** perspective, we propose a **novel adaptive hypergraph expansion method** to convert hypergraphs into weighted graphs, which is designed as a general framework that can be used over all hypergraphs and can be further learned for different downstream tasks, such as node classification, hyperedge prediction, and community detection. I believe our model AdE provides new insights into hypergraph learning.
>
> \
> \
> [1] You are AllSet: A Multiset Function Framework for Hypergraph Neural Networks, ICLR 2022.
>
> [2] HyperGCN: A New Method of Training Graph Convolutional Networks on Hypergraphs, NeuIPS 2019.

---

> ### Author Response · Authors · 2023-11-20
> **Response to Follow-up Comments of Reviewer qSDB (2/3)**
>
> *$\textbf{Q2:}$  Why are 2 selected nodes good for preserving hyperedge information? Why not select 3 nodes or, in particular, an adaptive number of nodes?*
>
> $\textbf{A2:}$ We are so sorry for the misleading citations. Citations 3 and 4 in the previous response introduce the definitions of bar-bell graphs. We were supposed to cite [3,4] in the current response for clearer clarification. In Section 2 of **[3]** (page 6), this work **demonstrates** that $\text{max}\_{v_i,v_j\in e}|\mathbf{S}_i - \mathbf{S}_j|$ r**epresents the most essential piece of information about the corresponding hyperedge $e$**. Therefore, this work defines the Hypergraph Markov Operator in Definition 2.1 which also selects two representative nodes to represent each hyperedge to analyze the hyperedge structure. Please refer to [3,4] for detailed proof. Based on the mentioned proof, we also leverage the idea of selecting two representative nodes to represent the information within the hyperedges. The reason why existing works [2,3,4] select two representative nodes based on the rule $(v\_{e^+}, v\_{e^-})=\arg\max\_{v_i,v_j\in e}|\mathbf{S}_i - \mathbf{S}_j|$ is that **finding the largest node and the smallest node in the feature space is an easy but effective way to represent the most essential information within the hyperedge**. However, suppose we would like to select three or more representative nodes for the corresponding hyperedge, it could be challenging and hard to find suitable ones.
> \
> \
> Next, we would like to explain with an **intuitive example**. Assuming that the hyperedge is a segment on a one-dimensional coordinate, with its nodes lying along this segment. The line segment formed by the minimum and maximum nodes on the coordinate axis can best encompass the information of this hyperedge. If we consider the hyperedge to be part of a two-dimensional coordinate, with its nodes falling within this region. The portion between the angles formed by the minimum point and the coordinate axis, as well as the maximum point and the coordinate axis, can cover all the nodes on the hyperedge. In other words, it can effectively retain the information of the hyperedge to the maximum extent. These explanations apply similarly in higher-dimensional spaces. However, we believe how three or more nodes in these coordinates represent the information within hyperedge is a challenging but worthwhile question for future exploration.
>
> *$\textbf{Q3:}$ Not sure whether the contribution of AdE is incremental, especially considering that its performance does not have much advantage over the baselines.*
>
> $\textbf{A3:}$ I understand your concerns but we believe our model is rational, novel, and solid in exploring hypergraph learning. Next, in order to alleviate your concerns,  we would like to discuss the contribution of this work from the following two aspects:
>  1. We point out the limitations of the existing expansion methods, i.e., clique-based expansion, star-based expansion, line-based expansion, and especially the expansion method in HyperGCN.  Based on the limitations of non-representative node pairs and fixed edge weights in HyperGCN, we propose a **novel model** including the adaptive node pair selection and the distance-aware kernel function novel for edge weight to enhance the hypergraph expansions. These two modules are specifically designed to address the proposed limitations, which are novel and solid with sufficient justifications and comprehensive experiments, including ablation study, complexity analysis, and intuitive examples. We believe our designed **AdE will provide sufficient insights into future hypergraph learning**.
>
>
>  2. We would like to demonstrate the effectiveness of AdE **empirically**. In Table 1 of the revised manuscript, AdE outperforms all existing expansion methods (i.e., CE-based, SE-based, LE-based, and Uni-based). From Table 2 in revision, we find out that AdE outperforms all baseline methods over most of the datasets. Specifically, Table 1 in **A1**  of the current-round responses again shows the mean improvement of the SOTA AllSet, HyperGCN, and our method AdE. We find out that **SOTA AllSet **gains **0.42 improvement** on average compared with the best baseline method, and the mean improvement of **AdE is 0.73** compared with the best baseline methods. Although the improvement seems to be not that significant over some datasets, we strongly believe that it is still a comparable improvement in hypergraph learning compared with SOTAs.  Moreover, we believe there is still room for hypergraph expansion and we will devote ourselves to exploring hypergraph learning in future research.
>
> \
> \
> [2] HyperGCN: A New Method of Training Graph Convolutional Networks on Hypergraphs, NeuIPS 2019.
>
> [3] Hypergraph Markov operators, eigenvalues, and approximation algorithms, STOC 2015.
>
> [4] Re-revisiting learning on hypergraphs: Confidence interval and subgradient method, ICML 2017.

---

> ### Author Response · Authors · 2023-11-23
> **Response to Third-round Comments of Reviewer qSDB (1/3)**
>
> We really appreciate your efforts in reading our response carefully. According to all the concerns you mentioned above, we will explain them one by one as follows:
>
> $\textbf{Q1:}$ The significance of the proposed method's empirical performance still seriously lacks justification. My concern that "the improvement over the best baselines is within one standard deviation" has not been addressed at all in the response. The authors have gone to great lengths to establish that "Allset or HyperGCN does not outperform baselines very much.” I truly appreciate their effort. However, this does not help at all to justify the empirical significance of the proposed method.
>
> \
> $\textbf{A1:}$ Thank you for your comments on the empirical experimental results. However, we believe that significant performance improvement should **not** be **the only criterion** to evaluate the contribution of a work. In this work, we can regard HyperGCN as our backbone, we list **Table 1** to compare HyperGCN and AdE. From this table, we can easily find out that our proposed method **AdE outperforms HyperGCN significantly** ( **2.82 on average over seven datasets**), showing that our proposed method is **effective and solid** enough. Compared with SOTA AllSet, we outperform AllSet over most (six out of seven) of datasets and gain comparable performance on the DBLP dataset. Even SOTA AllSet fails to outperform the best baseline over some datasets (eight out of fifteen). We do not think the statement *The significance of the proposed method's empirical performance still seriously lacks justification.* is supportive and convincing based on all experimental results.
>
> \
> **Table 1**: Performance comparison between backbound HyperGCN and AdE.
>
> | |Cora-CA | DBLP | Cora | Citeseer |Pubmed| House (1) | House (0.6) |
> |---|---|---|---|---|---|---|---|
> | HyperGCN | 79.08 ± 1.53 | 89.54 ± 2.93| 78.39 ± 2.03|70.39 ± 1.09|82.43 ± 1.88| 64.71 ± 1.71 | 79.63 ± 2.87|
> |AdE-GCN| $\textbf{82.07 ± 0.88}$|$\textbf{ 90.86 ± 0.16}$|$\textbf{80.74 ± 1.94}$|$\textbf{72.49 ± 0.94}$|$\textbf{87.88 ± 0.43}$| $\textbf{67.80  ± 1.50}$|$\textbf{82.06 ± 2.58}$ |
> |Improvement| 2.99   | 1.32   | 2.35   | 2.10   | 5.45   |3.09 | 2.43 |
> |||||||||
>
> On the other hand, we are unsure whether you totally understand our model design based on your follow-up response. We would like to emphasize our model design to facilitate your understanding of our model. Unlike existing hypergraph neural networks (e.g., AllSet and HGNN), we design a novel hypergraph expansion method to adaptively generate a weighted graph that can preserve the high-order relationship within the hypergraph to a large extent. In other words, **the generated weighted graph can be learned adaptively for different downstream tasks** (e.g., node classification and hyperedge link prediction), which **none of the existing expansion** methods (i.e., CE, SE, LE, and UE) can **achieve** that.  To achieve this, we design an adaptive and novel network GSi-Net to select adaptively select representative nodes for hyperedges. Besides,  to preserve the high-order relationship within hyperedge to a larger extent, we design a novel distance-aware kernel function to adaptively learn the edge weights between node pairs in our weighted graph. We are not sure whether you realize our novel distance-aware kernel function as you did not mention anything about the adaptive edge weights in your response.

---

> ### Author Response · Authors · 2023-11-23
> **Response to Third-round Comments of Reviewer qSDB (2/3)**
>
> *$\textbf{Q2:}$ I am still not convinced of the idea's novelty and technical depth. It has been clear that the proposed method still mostly follows the HyperGCN's backbone. That is, (1) selecting two representative nodes for each hyperedge, (2) doing expansion by connecting the two representative nodes with every other node in the hyperedge, and (3) running a GNN on the expansion. I do not think that the current refinement of the node selection process contributes enough novelty or technical depths that are suitable for publication at ICLR this time -- especially given that the performance improvement is so marginal according to Tables 1 and 2.*
> \
> \
> $\textbf{A2:}$ As we mentioned in $\textbf{A1}$, we are not quite sure whether you understand the main idea of this work comprehensively, and we would like to explain it as follows:
> \
> \
> *1) selecting two representative nodes for each hyperedge.*
> \
> \
> **A**: Yes. We first select two representative nodes for each hyperedge during hypergraph expansion. However, it is not an easy selection. We design a novel GSi-Net to select the representative two nodes adaptively. The selection process is learned during the model training for different downstream tasks. For example, the representative nodes will be different for node classification tasks and hyperedge link prediction tasks. And **the optimal structure of the weighted graph will be learned for different downstream tasks**. Therefore, our representative node selection process is aware of the downstream tasks.
>
> \
> \
> *(2) doing expansion by connecting the two representative nodes with every other node in the hyperedge.*
> \
> \
> **A**: Yes and No. First of all, the rest of the nodes (mediators) will connect two representative nodes, respectively. However, you miss the process of how we generate edges between node pairs and how we learn the edge weights for node pairs, which is another novelty of our model design. We design a **novel distance-aware kernel function** to learn the edge weights dynamically. **The edge weight between node pairs will also be different if we handle different downstream tasks**, which is common in real-world scenarios. Designing an adaptive hypergraph expansion method can handle various downstream tasks in a comprehensive way.
>
> \
> \
> *(3) running a GNN on the expansion.*
> \
> \
> **A**: Yes. We need to mention that our expansion method is designed as a general expansion framework that can handle various hypergraphs. With the adaptive weighted graphs, our method is applicable to any GNN method over different downstream tasks  (e.g., node classification, hyperedge prediction, and community detection), showing **its strong applicability** in real-world scenarios.
>
> \
> \
> *I do not think that the current refinement of the node selection process contributes enough novelty or technical depths that are suitable for publication at ICLR this time -- especially given that the performance improvement is so marginal according to Tables 1 and 2.*
> \
> \
> **A**:  We understand your concerns but we hold different opinions on that. We propose a novel hypergraph expansion method to generate adaptive weighted graphs for different downstream tasks. We also conducted experiments on the hyperedge link prediction task in the Appendix. We strongly believe **AdE is the first work to expand hypergraphs into adaptive weighted graphs for different downstream tasks**.
>
> \
> \
> *$\textbf{Q3:}$ My concern about "why always selecting two representative nodes" is partially resolved. However, I do believe that the presentation of this paper should be significantly refined to provide a principled introduction to this idea. It just seems too natural that for large hyperedges using three or four representative nodes, instead of two, can be beneficial to performance. However, this potential was never discussed and experimented.*
>
>
> $\textbf{A3:}$ We appreciate your constructive suggestion. However, we believe that selecting two representative nodes to represent the information within the hyperedge is a **common idea for hypergraph learning** [1-4]. For instance, HyperGCN [1] chooses two nodes for each hyperedge for reconstruction. In these works [2-4], the authors also select two nodes by default for each hyperedge and propose a diffusion method based on the weighted graph. None of the explanations about two representative nodes are provided in these works or are required to be discussed as the idea seems to be easy and common. We would like to discuss more about two representative nodes in our revision and we will explore more into this further in our future research.
>
> \
> [1] HyperGCN: A New Method of Training Graph Convolutional Networks on Hypergraphs, NeuIPS 2019.
>
> [2] Generalizing the Hypergraph Laplacian via a Diffusion Process with Mediators. COCOON 2019.
>
> [3] Spectral Properties of Hypergraph Laplacian and Approximation Algorithms, J. ACM 2018.
>
> [4] Hypergraph Markov Operators, Eigenvalues and Approximation Algorithms, STOC 2015.

---

### Author Response · Authors · 2023-11-17
**Kind Reminder to All Reviewers**

Dear All Reviewers:

We appreciate all of your valuable comments and suggestions. We have clarified all of your questions and concerns in each response and also revised some statements in our revisions, which are emphasized in blue.

We are glad that all reviewers agree the model design is novel in hypergraph learning. Most questions are related to model details, baseline comparison, and data statistics (Q1, Q2, Q3, and Q5 of Reviewer qSDB, Q4 of Reviewer UEpe, Q2 and Q3 of Reviewer 1ZMR, and Q1, Q2, and Q4 of Reviewer 7Lez), which we have thoroughly explained in each response.

To further show the strong applicability of AdE, we apply AdE for the hyperedge link prediction task during the rebuttal (**Appendix G**). Besides, we also discuss existing works about node-degree preserving hypergraph projection and further conduct another set of experiments to compare AdE and IRMM (**Appendix H**). Both additional experiments again demonstrate the effectiveness and generalization of AdE. Moreover, we list an intuitive example to show how AdE works in real-world scenarios (**Appendix I**).

To conclude, this work proposes a novel hypergraph expansion AdE to address the existing challenges in hypergraph expansion. Sufficient theoretical justification and comprehensive empirical experiments demonstrate the rationality, effectiveness, and generalization of AdE. We sincerely hope that our responses have adequately addressed all of your concerns, and please do let us know if you have any further comments or questions.

---

### Meta-Review · Area_Chair_jBa3 · 2023-12-09

**Metareview:**

This paper went over a significant discussion between authors and reviewers. The paper proposes an adaptive hypergraph expansion method to first select two representative nodes in a hyperedge based on their scalar feature projection differences, and then adaptively constructs weighted edges between the representative nodes and other nodes based on feature similarity. Based on the expanded graph, they run GNN to perform the node classification tasks. After reading all the reviews and discussions, I agree with the first and fourth reviewer that the proposed method still has unaddressed limitations preventing its acceptance. Firstly, as pointed out by multiple reviewers, the improvement over baselines is marginal, although the improvement is consistent. This is not a big issue for a theoretical paper, but a more significant improvement is expected for an empirical paper like this with many heuristic choices. Secondly, many choices are heuristic with only intuitive explanations, e.g., selecting 2 instead of adaptive number of representative nodes, the particular expansion method resulting in the bar-bell shape graph, and using distances between node features to construct weighted edges (which strongly relies on homophily of the graph). These choices lack formal theoretical justifications, with the provided propositions somewhat straightforward. Thirdly, the underlying homphily assumption is very important for the method to work but is not discussed. I expect to see more experiments on heterophily graphs. It does not need to also perform well on heterophily graphs, but properly showing a method's limitations and suit cases is desired.

**Justification For Why Not Higher Score:**

Many heuristic choices with no theoretical justifications. Heavily rely on homophily assumption for the method to work.

**Justification For Why Not Lower Score:**

N/A

---

### Decision · Program_Chairs · 2024-01-16

Reject